# Identifying regulators of associative learning using a protein-labelling approach in *Caenorhabditis elegans*

**Aelon Rahmani[1], Anna McMillen[1], Ericka Allen[1], Radwan Ansaar[1], Renee Green[1], Michaela E Johnson[1], Anne Poljak[2], Yee Lian Chew[1]***

[1]Flinders Health and Medical Research Institute, College of Medicine and Public Health, Flinders University, Adelaide, Australia; [2]Bioanalytical Mass Spectrometry Facility, Mark Wainwright Analytical Centre, University of New South Wales, Sydney, Australia

## eLife Assessment

In reporting on a **valuable** "learning proteome" for a *C. elegans* gustatory associative learning paradigm, this work identifies a new set of genes to be tested for roles in learning and memory, describes molecular pathways involving these genes and relevant for learning and memory in *C. elegans*, and deliver a new set of tools for prodding worm behavior. The methods and results **convincingly** support the findings, which will be of interest to neuroscientists and developmental biologists seeking to understand the self-assembly and operation of neural circuits for learning and memory.
[Editors' note: this paper was reviewed by Review Commons.]

***For correspondence:**
yeelian.chew@flinders.edu.au

**Competing interest:** The authors declare that no competing interests exist.

**Abstract** The ability to learn and form memories is critical for animals to make choices that promote their survival. The biological processes underlying learning and memory are mediated by a variety of genes in the nervous system, acting at specific times during memory encoding, consolidation, and retrieval. Many studies have utilised candidate gene approaches or random mutagenesis screens in model animals to explore the key molecular drivers for learning and memory. We propose a complementary approach to identify this network of learning regulators: the proximity-labelling tool TurboID, which promiscuously biotinylates neighbouring proteins, to snapshot the proteomic profile of neurons during learning. To do this, we expressed the TurboID enzyme in the entire nervous system of *Caenorhabditis elegans* and exposed animals to biotin only during the training step of an appetitive gustatory learning paradigm. Our approach revealed hundreds of proteins specific to 'trained' worms, including components of molecular pathways previously implicated in memory in multiple species such as insulin signalling, G-protein-coupled receptor signalling, and MAP kinase signalling. Most (87–95%) of the proteins identified are neuronal, with relatively high representation for neuron classes involved in locomotion and learning. We validated several novel regulators of learning, including cholinergic receptors (ACC-1, ACC-3, LGC-46) and putative arginine kinase F46H5.3. These previously uncharacterised learning regulators all showed a clear impact on appetitive gustatory learning, with F46H5.3 showing an additional effect on aversive gustatory memory. Overall, we show that proximity labelling can be used in the brain of a small animal as a feasible and effective method to advance our knowledge on the biology of learning.

## Introduction

All animals with a brain have the capacity to change their behaviour in response to changes in the environment. This capacity – to learn and remember – is essential for survival. There are numerous structural and molecular changes in the brain that modulate learning and memory in specific brain regions, occurring in a time and context-dependent manner (examples in *Huckleberry et al., 2016*; *Lin et al., 2010*; *Peixoto et al., 2015*; *Watteyne et al., 2020*; reviewed in *Bailey et al., 2015*). Research using model organisms has been essential towards understanding the key regulatory mechanisms underlying learning, many of which involve neurotransmitter signalling, neuromodulator signalling, signal transduction pathways, and cytoskeletal dynamics (*Rahmani and Chew, 2021*; *Peng et al., 2011*; *Lamprecht, 2014*). Importantly, many of these mechanisms appear to be conserved across diverse species (*Bailey et al., 2015*; *Matsumoto et al., 2018*; *Rahmani and Chew, 2021*).

Multiple studies have demonstrated that changes in the neuronal proteome are required for learning and memory formation (*Inberg et al., 2013*; *Barzilai et al., 1989*; *Rosenberg et al., 2014*). New protein synthesis appears to be critical in several contexts, as the addition of a protein synthesis inhibitor (e.g. cycloheximide) has been shown to abolish long-term memory (*Chen et al., 2012*; *Pedreira et al., 1995*; *Hernandez and Abel, 2008*). Moreover, protein degradation together with new protein synthesis has been strongly implicated in synaptic plasticity and memory formation (*Lee et al., 2008*; *Fazeli et al., 1993*; *Park and Kaang, 2019*). There is also evidence that local translation in neurons, specifically the synthesis of specific proteins in dendritic regions (thereby altering local proteome composition), plays a key role in learning (*Bradshaw et al., 2003*; *Das et al., 2023*; *Smith et al., 2005*; *Sutton and Schuman, 2006*). Additionally, several key regulatory proteins have been shown to be required at specific timepoints, such as during the training/learning step, to trigger memory formation (*Stefanoska et al., 2023*; *Watteyne et al., 2020*). Taken together, these findings suggest that the spatiotemporal regulation of protein composition within neurons is critical for learning and memory formation.

The molecular requirements for learning have primarily been identified by combining genetic approaches with behavioural paradigms to test learnt associations, typically through assaying candidate genetic mutants or performing a forward genetics screen. These strategies have been extremely insightful; however, they have some limitations. The first being that candidate genetic screens are time-consuming and labour-intensive and require subjective selection of which candidate genes to test (for example, *Hukema, 2006*; *Stein and Murphy, 2014*). The second is that large-scale screens tend to only reveal genes that have the strongest phenotypes, so genes that have more subtle phenotypes (*Hiroki and Iino, 2022*; *Lindsay et al., 2022*), or act in redundant pathways (*Feng et al., 2010*; *Gyurkó et al., 2015*; *Shahmorad, 2015*), may not be identified using these approaches despite their contributions to learning.

To overcome these limitations, and to gain a holistic view of the molecular pathways that contribute to learning, we used an objective proteomics approach to snapshot the protein-level changes that occur specifically during learning. To do this, we expressed the proximity-labelling tool TurboID in the entire *Caenorhabditis elegans* nervous system and used this to identify the proteins present in neurons during the training step of an associative learning paradigm we call 'salt associative learning'. TurboID is an enzyme based on the BirA* biotin ligase, engineered to provide greater catalytic efficiency (*Branon et al., 2018*) compared with the original BirA* enzyme used in BioID experiments (*Roux et al., 2012*). TurboID catalyses a reaction where biotin is covalently added onto lysine residues – as this process requires biotin, its timing can be controlled by depleting tissues of biotin, then adding it exogenously only at specific time points. Additionally, spatial control can be provided by regulating the site of TurboID expression using cell-specific transgenes. TurboID has been used in multiple studies for identification of protein-protein interactions, usually by tagging a 'bait' protein N- or C-terminally with the TurboID enzyme, allowing for rapid biotinylation of bait interactors. Through this approach, TurboID has been used for protein-tagging experiments in *C. elegans* (*Artan et al., 2022*; *Sanchez et al., 2021*; *Holzer et al., 2021*; *Hiroki et al., 2022*). For example, this approach identified cytoskeletal proteins in *C. elegans* proximal to the microtubule-binding protein PTRN-1 (*Sanchez et al., 2021*), and detected interactors for ELKS-1, which localises other proteins to the presynaptic active zone in the nervous system (*Artan et al., 2021*).

In our study, rather than focusing on specific protein-protein interactions, we expressed TurboID that was not tagged with any bait protein in the entire nervous system of *C. elegans* to identify

as many proteins as possible within the cytoplasm. Using this approach, we identified hundreds of proteins specific to 'trained' worms, which we refer to here as the *learning proteome*, including those in molecular pathways previously shown to contribute to learning and memory formation in worms and other organisms. In addition, we validated several novel regulators of gustatory learning, including cholinergic receptors (ACC-1, ACC-3, and LGC-46), Protein kinase A regulator KIN-2, and putative arginine kinase F46H5.3. These proteins all show a clear impact on appetitive gustatory learning. F46H5.3 showed an additional effect on aversive gustatory learning, suggesting a more general role for this kinase in memory encoding. In summary, we have demonstrated that our approach to using proximity labelling to snapshot the brain of a small animal during training is a feasible and effective method to further our understanding of the biology of learning.

## Results

### TurboID expression in the nervous system of *C. elegans* successfully labels proteins during learning

To model learning in *C. elegans*, we used a simple yet robust associative learning paradigm called salt associative learning. Briefly, this assay involves training worms to associate the absence of salt (NaCl) with the presence of food. *C. elegans* is typically grown in the presence of salt (usually ~50 mM) and displays an attraction toward this concentration when assayed for chemotaxis behaviour on a salt gradient (*Kunitomo et al., 2013*; *Luo et al., 2014*). Training/conditioning with 'no salt +food' partially attenuates this attraction (group referred to 'trained'). This is because the presence of abundant food (unconditioned stimulus) is a strong innate attractive cue, and pairing this with 'no salt' (the conditioned stimulus) leads to the animals showing the same behaviour towards the conditioned stimulus as they do to the unconditioned stimulus, that is attraction towards no salt, reflected as a preference for lower salt concentrations (*Hiroki et al., 2022*; *Nagashima et al., 2019*). Similar behavioural paradigms involving pairings between salt/no salt and food/no food have been previously described in the literature (*Nagashima et al., 2019*). Here, learning experiments were performed by conditioning worms with either 'no salt +food' (referred to as 'salt associative learning') or 'salt +no food' (called 'salt aversive learning').

To identify the learning proteome, we adapted this learning paradigm to incorporate TurboID-catalysed biotinylation of proteins specifically during the learning/conditioning step. We did this by (1) performing the salt associative learning assay on transgenic animals expressing TurboID in the entire nervous system (P*rab-3::TurboID*) and (2) adding biotin only when the worms are being trained (i.e. exposed to both food and 'no salt' in the 'trained' group, or to food and 'high salt' concentrations in the 'high-salt control' group). As an additional control, we performed the same assay on non-transgenic (non-Tg) animals that do not express TurboID (*Figure 1A and B*). We then isolated proteins from >3000 whole worms per group for both 'high-salt control' and 'trained' groups, most of which were subjected to a sample preparation pipeline for mass spectrometry, and some of which were probed via western blotting to confirm the presence of biotinylated proteins. The same pipeline was used to generate five biological replicates.

Validation of TurboID-catalysed biotinylation was performed in two ways: First, we compared total protein from naïve/untrained animals that are non-Tg versus TurboID-encoding by western blot and probed for V5-tagged TurboID: as expected, we observed expression in transgenic worms only at the predicted size (39 kDa) (*Figure 1C*). Secondly, we tested if exposure to biotin increased the biotinylation signal in a TurboID-dependent manner. To do this, we quantified the biotinylation signal in (1) naive non-Tg worms not exposed to biotin, (2) non-Tg *C. elegans* exposed to biotin for 6 hr, (3) naive TurboID worms not exposed to biotin, and (4) TurboID animals exposed to biotin for 6 hr. Although background biotinylation was present in worms not treated with biotin, we found that biotin exposure increased the signal 1.3-fold for non-Tg and 1.7-fold for TurboID *C. elegans* (*Figure 1—figure supplement 1*). Taken together, these findings indicate that there is increased biotinylation of proteins in the presence of both biotin and the TurboID enzyme.

Mass spectrometry experiments were performed with the following experimental groups per replicate: (1) non-transgenic/non-Tg high-salt control, (2) non-Tg trained, (3) TurboID high-salt control, and (4) TurboID trained. We did not include no-biotin treatment controls due to the practical challenges of handling >4 groups in the combined learning assay/mass spectrometry pipeline, for which >3000

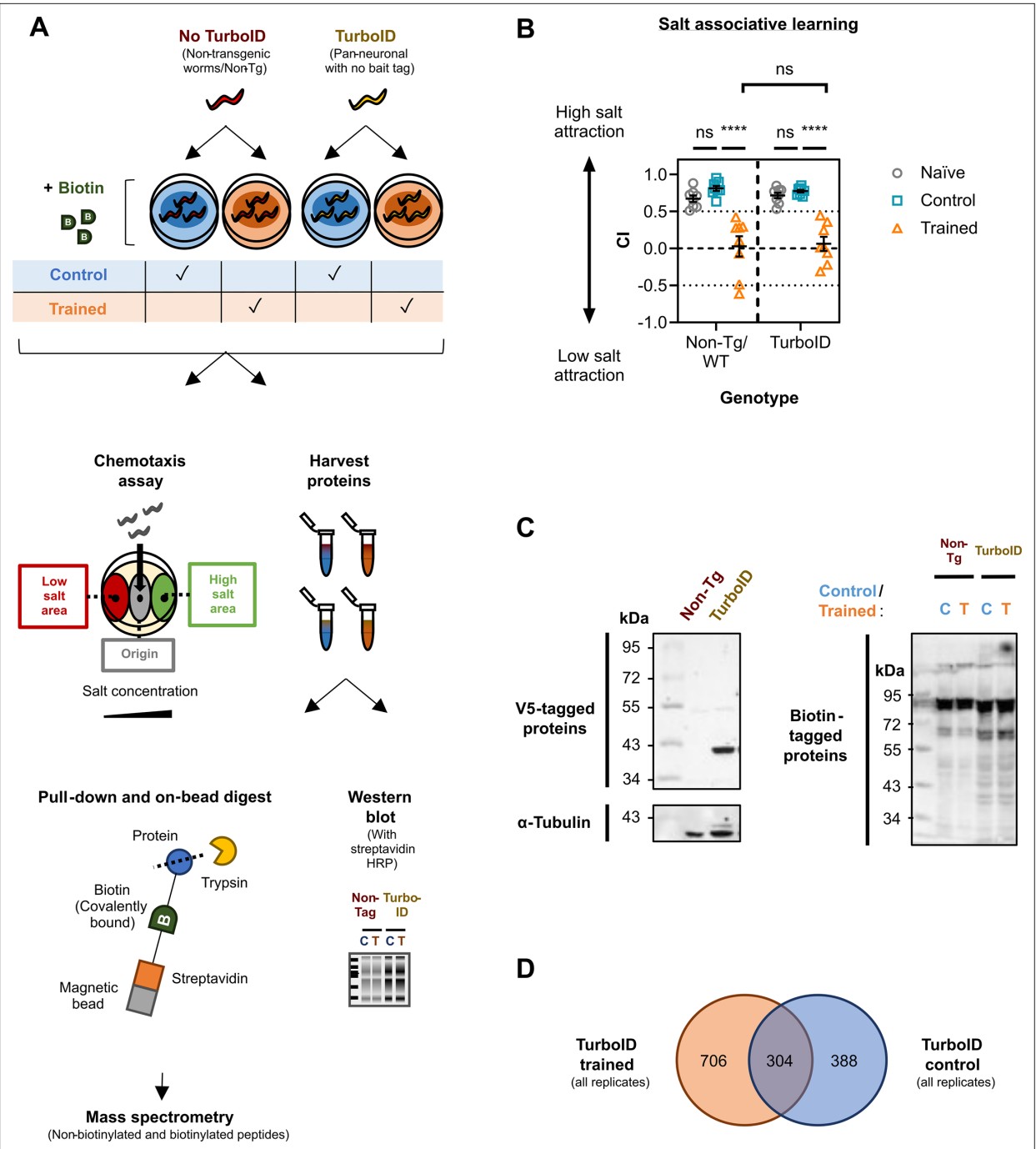

**Figure 1.** Summary of the TurboID approach for protein labelling in all *C. elegans* neurons during learning. (**A**) Workflow for mass spectrometry-based analysis. Biotin-depleted animals without (non-transgenic/Non-Tg, red) or with TurboID (transgenic, yellow) were exposed to 1 mM of exogenous biotin during conditioning by pairing food with 'no salt' (orange - trained) or 'high salt' (blue - control). >3000 worms were used per group /biological replicate (n=5) – a small proportion of each group was tested in a chemotaxis assay to assess learning capacity, while the rest was subjected to sample preparation steps for mass spectrometry (see Materials and methods). Some harvested protein was probed via western blot for the presence of biotinylated proteins or V5-tagged TurboID (see panel C for representative image from replicate 1). (**B**) The graph shows chemotaxis assay data for Non-Tg/wild type (WT) and transgenic *C. elegans* following salt associative learning. Each data point represents a 'chemotaxis index' (CI) value for one biological replicate (n=8). Each biological replicate includes three technical replicates (26–260 worms/technical replicate). *Statistical analysis*: Two-way ANOVA and Tukey's multiple comparisons test (****≤0.0001; ns = non-significant). Error bars = mean ± SEM. (**C**) Western blots to visualise V5-tagged TurboID and biotinylated proteins. The left side shows V5-tagged TurboID visualised using 18 µg total protein from naïve worms per lane (39 kDa). Non-Tg protein lysates acted as a negative control. α tubulin was probed as a loading control. The right side shows biotinylated proteins visualised from 25 µg total protein per lane from control (**C**) or trained (**T**) worms with streptavidin-horseradish peroxidase (HRP). (**D**) Venn diagram comparing

*Figure 1 continued on next page*

*Figure 1 continued*

all proteins assigned an identity by MASCOT from peptides detected by mass spectrometry from transgenic worms. Values represent the number of proteins listed as detected in 'TurboID, control' (blue) and 'TurboID, trained' (orange). These lists were generated by first subtracting proteins identified in corresponding Non-Tg lists and then comparing both control and trained TurboID lists. The overlap represents proteins unique to 'TurboID, trained' worms in ≥1 replicate/s that were also detected in 'TurboID, control' worms in ≥1 other replicate/s.

The online version of this article includes the following source data and figure supplement(s) for figure 1:

**Source data 1.** Original western blot membrane images corresponding to *Figure 1C*.

**Source data 2.** Raw data corresponding to *Figure 1C*.

**Figure supplement 1.** Western blot to quantify biotin-tagged protein levels following biotin exposure.

**Figure supplement 1—source data 1.** Original western blot membrane images corresponding to *Figure 1—figure supplement 1*.

**Figure supplement 1—source data 2.** Raw data corresponding to *Figure 1—figure supplement 1*.

**Figure supplement 2.** Design of chemotaxis assay plates to quantify salt chemotaxis behaviours.

**Figure supplement 3.** Western blots to assess biotinylation in *C. elegans* by TurboID during memory encoding of salt associative learning.

**Figure supplement 3—source data 1.** Original western blot membrane images corresponding to *Figure 1—figure supplement 3*.

**Figure supplement 3—source data 2.** Raw data corresponding to *Figure 1—figure supplement 3*.

**Figure supplement 4.** Assessing overlap for proteins detected in trained worms across all biological replicates.

worms are required per group. Therefore, all groups were exposed to biotin during the 6 hr exposure period to food and either high salt (for control) or no salt (for trained; *Figure 1A*).

To confirm that each experimental group displayed the expected phenotype after training, a portion of worms from all groups was tested using a chemotaxis assay. The chemotaxis index (CI) was used as a readout of learning performance: a positive CI reflects high salt preference, a CI close to 0 represents a more neutral response, and a negative CI represents low salt preference (*Figure 1— figure supplement 2*). We confirmed after each learning assay that naïve/untrained worms had a strongly positive CI (~0.7–0.9), whereas trained animals showed a lower CI (~0.0). We also performed a learning control (indicated as high-salt 'control') in which the presence of food (the US) is paired with high salt concentrations – worms in this group are attracted to high salt and showed a strongly positive CI (~0.7–0.9; *Figure 1B*), displaying a similar behaviour to naïve worms. This behavioural change seen in trained animals, versus the naïve and high-salt control groups, represented successful learning as seen in previous studies (*Hiroki et al., 2022*; *Nagashima et al., 2019*). This was observed in both non-Tg and transgenic animals, confirming that introducing the transgene did not perturb learning (*Figure 1B*).

We also confirmed by western blotting that biotinylated proteins could be observed in TurboID-expressing high-salt control and trained groups (*Figure 1C*). As in other *C. elegans* studies utilising TurboID, we saw background biotinylation in non-Tg controls; however, this is visually lower compared with groups from TurboID transgenic worms (*Figure 1C*; *Artan et al., 2021*; *Sanchez et al., 2021*). Quantification of the signal within entire lanes showed a 1.1-fold increase in the 'TurboID, control' lane compared with the 'non-Tg, control' lane, and a 1.9-fold increase in the 'TurboID, trained' lane compared with the 'non-Tg, trained' lane. For all replicates, we determined that biotinylated proteins could be observed from total TurboID-positive worm lysate by western blotting before proceeding with downstream proteomic experiments (*Figure 1—figure supplement 3*, *Supplementary file 1B*).

Our sample preparation methodology for mass spectrometry is based on similar protocols used in *C. elegans* and other systems (*Artan et al., 2022*; *Sanchez et al., 2021*; *Prikas et al., 2020*). We performed five biological replicates, in line with other *C. elegans* studies (*Artan et al., 2022*; *Holzer et al., 2021*). To examine the learning proteome, we first subtracted proteins from 'TurboID, trained' groups also present in 'Non-Tg, trained' samples to generate a protein list specific to 'TurboID, trained' animals for each biological replicate. We next subtracted from 'TurboID, control' lists any proteins that appeared in 'Non-Tg, control' samples to generate a revised 'TurboID, control' protein list specific to each replicate. We then compared revised protein lists for 'trained' and 'control' worms from all biological replicates and examined both unique and shared proteins between these two groups. We found 304 proteins that were shared between 'trained' and 'control' TurboID groups, 706 proteins unique to the 'TurboID, trained' group, and 388 proteins unique to the 'TurboID, control' group (*Figure 1D*). We refer to the *learning proteome* as proteins unique to

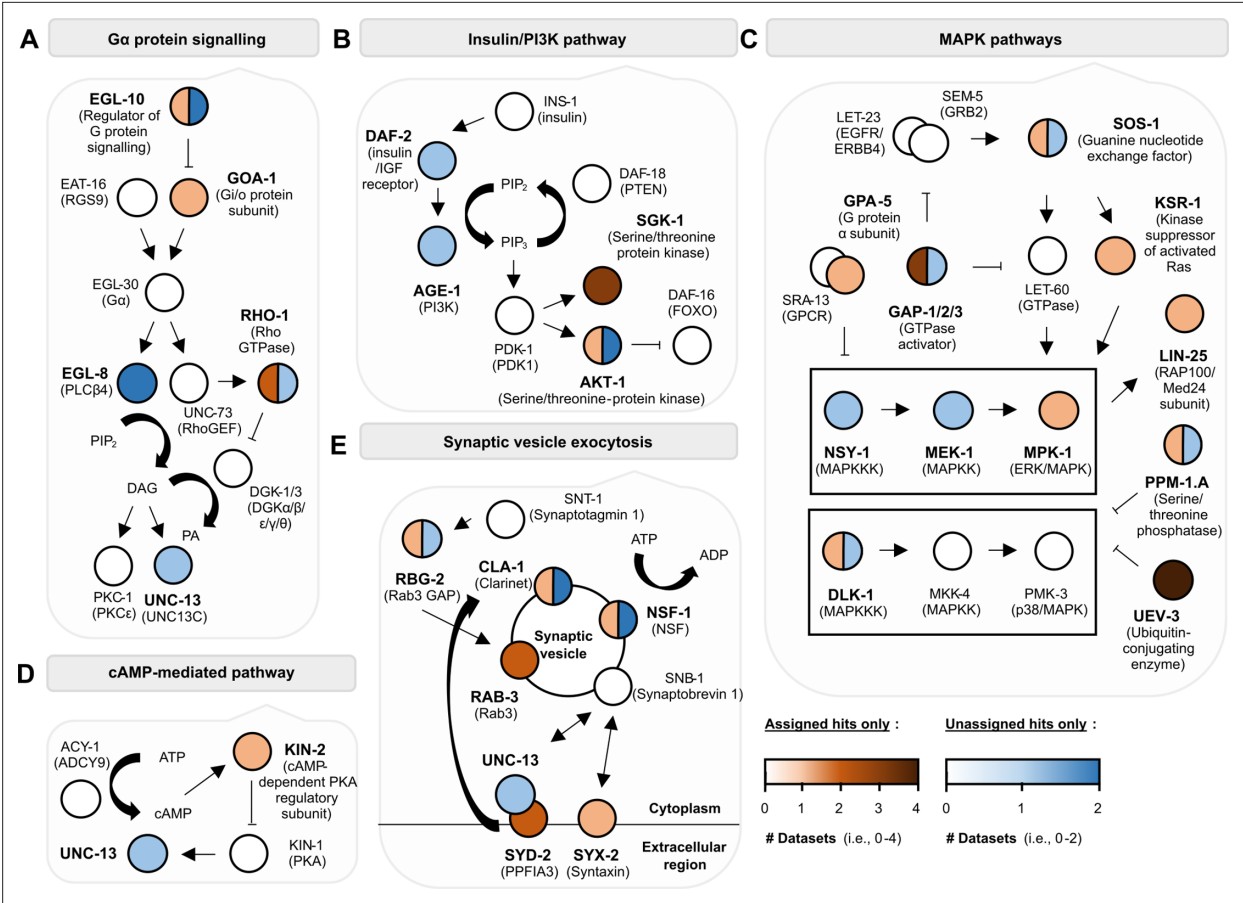

**Figure 2.** Molecular pathways previously implicated in associative learning are detected in our learning proteome. Proteins detected from 'TurboID, trained' worm lysates by mass spectrometry are in bold with circles coloured as orange ('assigned hits' assigned protein identities by MASCOT) and/or blue ('unassigned hits' given protein identities by bulk BLAST searching, but not MASCOT). Darker colours mean the protein has been detected in more than one biological replicate (see legend).

The online version of this article includes the following figure supplement(s) for figure 2:

**Figure supplement 1.** Analysis of molecular functions of assigned hits in the learning proteome.

samples for 'TurboID, trained' worms. When generating the learning proteome, we categorised proteins as 'assigned hits' based on the criteria that at least one unique peptide was identified by the MASCOT search engine for the protein identity from at least one biological replicate. We also examined peptide sequences in our peak lists that were considered 'unassigned' by MASCOT, as these sequences were not detected as unique for any protein by the software, but specific protein identities could be found by performing a Basic Local Alignment Search Tool (BLAST) query (https://blast.ncbi.nlm.nih.gov/Blast.cgi; see Materials and methods for details). The Venn diagram in *Figure 1D* shows assigned hits only. Learning proteome lists for both assigned and unassigned hits are in *Supplementary file 1C and D*.

We assessed overlap between biological replicates for individual candidates (*Figure 1—figure supplement 4*) using two mass spectrometry systems: Thermo-Fisher Q-Exactive Orbitrap ('QE') and Orbitrap Exploris ('Exploris'). Candidates detected in multiple replicates comprised 17% of assigned hits in QE runs, 13% in Exploris, and 21–23% when including unassigned hits (*Figure 1—figure supplement 4A–D*). Of the 1,010 assigned QE hits, 17% were also identified with Exploris, increasing to 29% when including all 2065 protein identities (*Figure 1—figure supplement 4E–F*). Despite modest overlap (<25%), key learning-related pathways (*Figure 2*, *Supplementary file 1*) and other biological processes, including metabolic pathways (*Figure 3*), were consistently represented, supporting the biological relevance of the identified learning proteome.

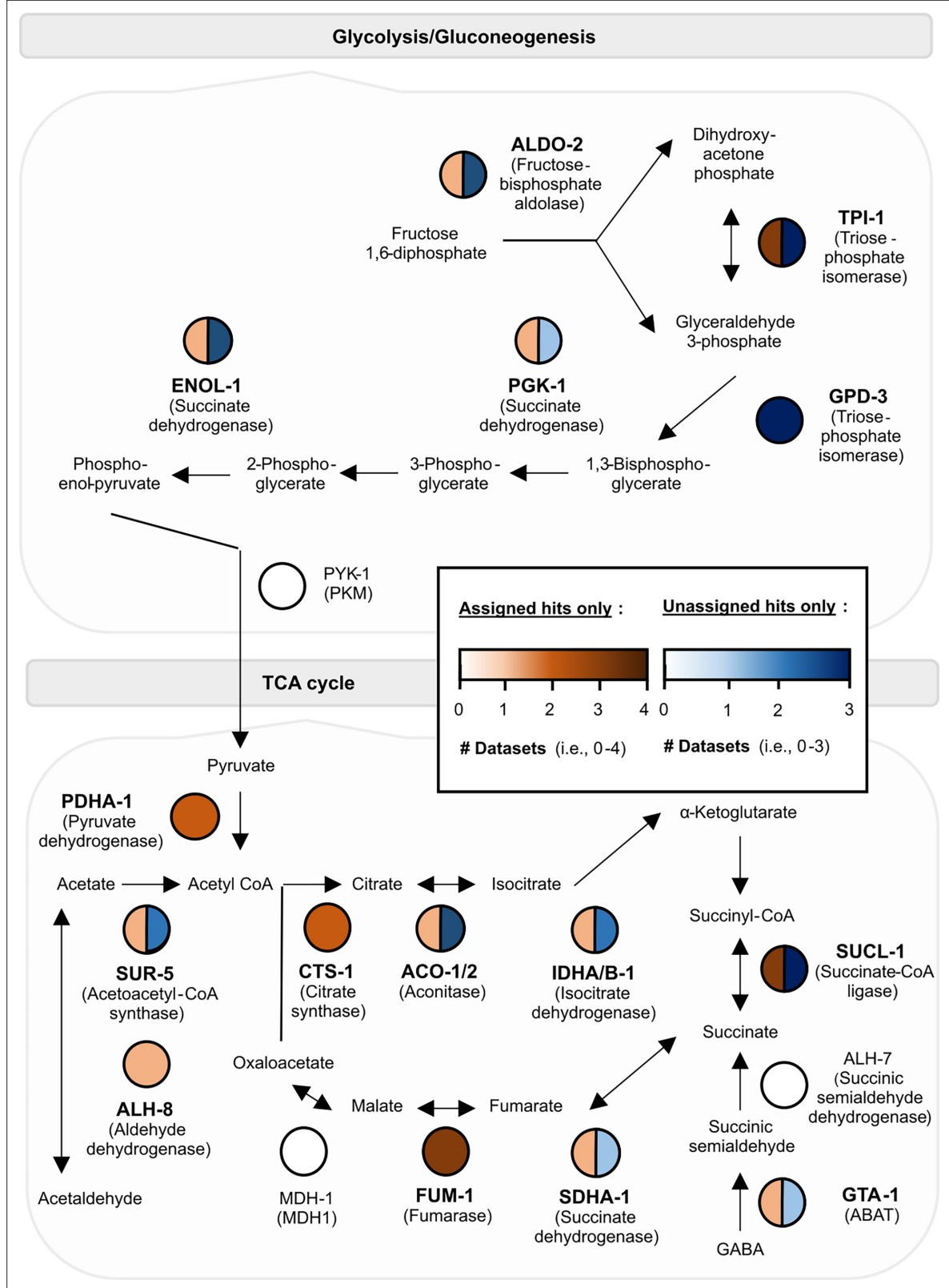

**Figure 3.** Schematics for metabolic processes represented in the learning proteome. The molecular pathways above are (**A**) carbohydrate metabolism (glycolysis and gluconeogenesis) and (**B**) fatty acid metabolism (via the tricarboxylic acid or TCA cycle). Each protein is a node in white (not detected by TurboID during learning), orange (an 'assigned hit'), and/or blue (an 'unassigned hit') based on mass spectrometry data from 'TurboID; trained' worms. Darker colours mean the protein has been detected in more than one biological replicate (see legend).

## Examination of the learning proteome reveals known regulators of learning and memory

Our initial analysis of the learning proteome sought to validate our TurboID-based approach by identifying components of biological pathways previously implicated in learning. We then performed a gene ontology (GO) term analysis of 'cellular component' to obtain a broad overview of the subcellular localisation of proteins identified in trained animals (*Figure 4—figure supplement 1A*). To do this, we generated protein-protein interaction (PPI) networks of assigned protein hits within the learning proteome for subcellular components of interest (*Figure 4—figure supplement 1B–G*), using data from STRING and curated with the Cytoscape ClueGO tool (*Supplementary file 1E and F*; *Bindea et al., 2009*). We found that a majority of proteins were categorised as 'cytoplasmic' (28.1%) as expected from our approach, which utilised the TurboID enzyme not tagged to any bait protein; this means that we would anticipate the enzyme to be present relatively evenly across the cell body and to catalyse biotinylation of proteins in this space. We saw that an unexpectedly high proportion of proteins were nuclear (18.1%), despite the presence of a nuclear export signal in our TurboID transgene, which should prevent TurboID from entering the nucleus and biotinylating nuclear proteins – this could be due to some 'leaky' entry of the enzyme or biotin reactive species into the nucleus, or that some proteins are categorised solely as 'nuclear' in the ClueGO database when they are also present in other cellular components. Importantly, we found that a proportion of proteins are categorised as present in neuronal compartments – the pre-synapse (0.5%), cilia/dendrites (2.7%), and in the axon/(synaptic) vesicles (4.0%) – as expected from transgenic expression of TurboID in the nervous system.

Learning and memory formation in organisms with brains of varying sizes has been shown to involve key regulatory pathways including signalling via neurotransmitters/neuromodulators, G-protein-coupled receptors (GPCR), the mitogen-activated protein kinase (MAPK) pathway, and the insulin/insulin growth factor-like pathway (*Matsumoto et al., 2018*; *Myhrer, 2003*; *Rahmani and Chew, 2021*). We next categorised proteins within the learning proteome (*Figure 2*) based on their known roles within these signalling pathways. This included (1) several GPCR components, including the $G_{i/o}$ protein subunit GOA-1 and $G_\alpha$ protein subunit GPA-2, (2) regulators of insulin signalling including the DAF-2 insulin receptor, phosphoinositide 3-kinase AGE-1, and serine/threonine protein kinases AKT-1 and SGK-1, which were previously reported to modulate salt-based learning in the worm (*Tomioka et al., 2006*; *Sakai et al., 2017*), (3) MAPK signalling components including NSY-1/MAPKKK, MEK-2/MAPKK, and MPK-1/MAPK/ERK, (4) cAMP/PKA (protein kinase A) signalling regulators such as the regulatory PKA subunit KIN-2, and (5) multiple components that modulate synaptic vesicle release, including N-ethyl-maleimide sensitive fusion protein NSF1/NSF-1 and syntaxin/SYX-2. In addition, we identified several proteins relevant to glutamate, acetylcholine, and GABAergic signalling. Several components involved in protein synthesis and degradation were also detected in our learning proteome, in line with studies that suggest changes in total protein composition following memory formation (*Inberg et al., 2013*; *Barzilai et al., 1989*). These data are summarised in *Figure 2*, *Figure 2—figure supplement 1*, and *Supplementary file 1F*. In summary, the learning proteome includes both known learning regulators and potentially novel candidates that warrant further study. We focused on proteins functioning within these pathways of interest in our subsequent investigations (highlighted nodes in *Figure 2*, *Figure 2—figure supplement 1*, and *Figure 4—figure supplement 1*).

In addition, we consistently observed enrichment of two metabolic pathways, fatty acid metabolism via the TCA cycle and carbohydrate metabolism (gluconeogenesis and glycolysis), in multiple biological replicates of mass spectrometry data, uniquely in TurboID-trained animals (*Figure 3*). These metabolic pathways play essential roles, including in cellular energy production and macromolecule biosynthesis (*Krebs and Johnson, 1937*; *Goetsch and Lu, 1993*). Consequently, their disruption can severely impair animal health. For example, knock-down of mitochondrial components involved in the TCA cycle led to larval arrest and/or severely reduced lifespan in *C. elegans* (*Artan et al., 2022*; *Liao et al., 2022*). This limits the capacity to assess these processes in learning using single-gene mutants or knockdown tools. Therefore, our TurboID approach reveals biological pathways potentially involved in memory formation that are not detectable through conventional forward or reverse genetic screens.

**Table 1.** Neuron-specific expression within the learning proteome.

Mass spectrometry runs (n=5) were performed with the ThermoFisher Scientific Q-Exactive Orbitrap ('QE') and/or ThermoFisher Scientific Orbitrap Exploris ('Exploris'), for technical reasons. There are six lists because replicate #3 was run on both mass spectrometers: corresponding protein lists are annotated as '3 a' and '3b', respectively. The total '#Assigned hits' versus '#All hits' (assigned +unassigned hits) is shown in rows listed above. The CeNGEN database (threshold = 2) was used to determine corresponding percentages for assigned hits versus all hits as '% Neuronal for assigned hits' versus '% Neuronal for all hits' (*Taylor et al., 2021*). The average percentages across all replicates were 91% for assigned hits only versus 89% for all hits.

| Biological replicate | 1 | 2 | 3a | 3b | 4 | 5 |
|---|---|---|---|---|---|---|
| Mass spectrometer used | QE | QE | QE | Exploris | Exploris | Exploris |
| #Assigned hits | 364 | 159 | 97 | 237 | 202 | 274 |
| #All hits | 675 | 516 | 279 | 455 | 708 | 578 |
| % Neuronal for assigned hits | 93 | 91 | 95 | 91 | 87 | 91 |
| % Neuronal for all hits | 91 | 89 | 92 | 90 | 89 | 89 |

## Exploring neuron class representation within the learning proteome

Aside from identifying relevant biological networks, we also used data from the learning proteome to identify potential neuron classes involved in memory formation, using four databases. This included the Wormbase Tissue Enrichment Analysis (TEA) Tool (*Angeles-Albores et al., 2016*), based on Anatomy Ontology (AO) terms, and single-cell transcriptomics data from the *C. elegans* Neuronal Gene Expression Network (CeNGEN; *Taylor et al., 2021*).

Firstly, we employed transcriptome databases to check representation of the nervous system within learning proteome data. The CeNGEN database confirmed that 87–95% of assigned hits and 89–92% of all hits (assigned and unassigned hits) from the learning proteome show neuronal expression, that is were found in at least one neuron in the database (*Table 1*). It is important to note that CeNGEN was generated using L4 hermaphrodites, and not young adult hermaphrodites (as used here for TurboID), since transcriptomes differ between the two developmental stages (*St Ange et al., 2024*). The nervous system transcriptome has been characterised for young adult hermaphrodites, highlighting 7873 genes that are enriched in neurons (versus non-neuronal tissues). Neuron-enriched genes were identified using single-nucleus and bulk neuron RNA-Seq techniques, respectively (*St Ange et al., 2024*; *Kaletsky et al., 2016*). The nervous system is highly represented by our proteome data; 75–87% of assigned hits and 75–83% of all hits correspond to neuron-enriched genes identified by St. Ange et al. and Kaletsky et al.

Secondly, we assessed which tissues and neuron classes are most highly represented within the learning proteome. We used the Wormbase TEA tool to search for gene lists corresponding to proteins encoded by (1) assigned hits only and (2) both assigned and unassigned hits within the learning proteome. Anatomical terms were considered enriched when they had a q value <1. We observed enriched terms for pharyngeal neurons (M1, M2, M5, NSM, and I4), sensory neurons (PVD), interneurons (ADA and RIG), ventral nerve cord (VNC) motor neurons (VB2, VB3, VB4, VB5, VB6, VB7, VB8, VB9, VB10, and VB11), and CAN cells from both gene lists. RIS interneurons and DD motor neurons were also enriched when including unassigned hits. Several of these neurons have previously been implicated in learning: RIG interneurons (*Zhou et al., 2023*) and NSM neurons in butanone olfactory learning (*Fadda et al., 2020*), VNC neurons through changes in glutamate receptor GLR-1 expression during touch habituation (*Rose et al., 2003*) and diacetyl aversive learning (*Vukojevic et al., 2012*), and RIS interneurons in salt aversive learning (*Wang et al., 2025*). Therefore, neurons enriched within the learning proteome include those known to be required for learning; other neurons not previously identified in this context, such as pharyngeal neurons, may warrant further study.

We complemented this analysis by using the CeNGEN database to search for gene lists encoding proteins (assigned hits only, minus non-transgenic controls) identified in control worms (388 genes) versus trained animals (706 genes) from *Figure 1D* (*Taylor et al., 2021*). Using the bulk gene search function in CenGEN (threshold = 2), we determined the number of genes from each list that are expressed in a specific neuron type. Values for the trained gene list were normalized to account for the ~1.8-fold increase in the number of proteins detected in trained samples compared to the

high-salt control. For each neuron class that appeared in both datasets (128 in total), we calculated fold-change values between the number of genes from trained vs control gene lists. Neurons were ranked in descending order of fold-change. This ranked list is based on the relative enrichment of training-associated genes compared to control, with higher ranks suggesting neurons that may be more transcriptionally responsive or involved during training. These data are summarised in *Supplementary file 1G*. Given that CeNGEN utilises transcriptomic data from L4 (juvenile) animals, neuron classes were also ranked using equivalent datasets for young adult hermaphrodites: Worm-Seq (*Ghaddar et al., 2023*) and CeSTAAN (*Princeton University, 2025*; see Materials and methods for details). Importantly, CeSTAAN and Worm-Seq provide data for 79 and 104 neuron classes, respectively (vs 128 from CeNGEN); this section therefore focuses on CeNGEN data due to its greater coverage, with other datasets described in brackets. Moreover, as this analysis is descriptive and does not include statistical testing (e.g. bootstrapping), the rankings should be interpreted as indicative rather than definitive, and future work incorporating formal statistical approaches will be important to validate these observations.

Cholinergic and glutamatergic neurons constituted 15% and 55% of neurons ranked #1–20, respectively (45% and 30% for CeSTAAN; 40% and 20% for Worm-Seq). Glutamate signalling components previously have been implicated in *C. elegans* learning paradigms involving salt (e.g. NMDA-type glutamate receptor subunits *nmr-1* and *nmr-2*; *Kano et al., 2008*). Acetylcholine has not been explored extensively in *C. elegans* for its involvement in learning but has been described in other animal models and in humans (reviewed in *Huang et al., 2022*). Other neuron classes identified have previously been implicated in salt-based associative learning (ranks in brackets): AVK interneurons (rank #7 for CenGEN; #37 for CeSTAAN; #76 for Worm-Seq; *Beets et al., 2012*), RIS interneurons (rank #14 for CenGEN; #1 for CeSTAAN; #31 for Worm-Seq; *Wang et al., 2025*), salt-sensing neuron ASEL (rank #18 for CenGEN; #16 for CeSTAAN; #34 for Worm-Seq as 'ASE'; *Beets et al., 2012*), CEP and ADE dopaminergic sensory neurons (individually ranked #22 and #39, respectively, for CenGEN, vs 'CEP_ADE_PDE' ranked #20 and #80 for CeSTAAN and Worm-Seq, respectively; *Voglis and Tavernarakis, 2008*), and AIB interneurons (rank #21 for CenGEN; #11 for CeSTAAN; #67 for Worm-Seq; *Sato et al., 2021*). In summary, although there are some exceptions that may be reflecting expression differences between adult and L4 animals, the same neuron classes are generally seen as highly represented for trained animals (vs control) for all three transcriptomic datasets.

Interestingly, unlike its counterpart ASEL (rank #18 for CenGEN), the salt-sensing neuron ASER was ranked only #104/128 (for CenGEN, *Supplementary file 1G*). ASER becomes activated in response to a decrease in salt concentration (*Suzuki et al., 2008*) and its downstream targets likely function to redirect worms toward higher salt concentrations (*Appleby, 2012*). This activation is suppressed after training that reduces attraction to high salt levels (*Sato et al., 2021*; *Wang et al., 2025*). It is possible that this learning-dependent suppression of ASER activity may explain its lower fold-change in trained versus control groups. Alternatively, given it is ranked #4 using the CeSTAAN database, potentially due to the use of adults and not L4, it is equally possible that additional proteomic changes may be required in ASER (vs ASEL, with rank #16) to trigger salt chemotaxis changes. Nevertheless, these findings imply a molecular and cellular switch facilitated by dual ASE neurons to express gustatory learning in the worm, which complements previous research.

Some neurons identified were not previously implicated in learning: IL1 polymodal head neuron class (rank #1 for CenGEN; #44 for CeSTAAN; #42 for Worm-Seq), motor neuron DA9 (rank #2 for CenGEN; #78 for CeSTAAN as 'PDA_AS_DA_DB'; #95 for Worm-Seq as 'DA_VA'), and interneuron DVC (rank #5 for CenGEN; #23 for CeSTAAN; #3 for Worm-Seq). IL1 releases glutamate (*Pereira et al., 2015*) and mainly functions in regulating foraging behaviour (*Hart et al., 1995*), potentially indicating a role in food-based responses. Separately, cholinergic neuron DA9 and glutamatergic neuron DVC are involved in backward locomotion (*Pereira et al., 2015*; *Ardiel and Rankin, 2015*; *Chalfie et al., 1985*). Changes in locomotion are critical for learning-dependent modulation of chemotaxis: the incidence of sharp turns or 'pirouette' movements in *C. elegans* is influenced by prior experience in salt-based gustatory learning (*Kunitomo et al., 2013*). IL1 may influence salt-based learning by signalling through interneurons AVE (rank #24 for CenGEN; #10 for CeSTAAN; #87 for Worm-Seq) and PVR (rank #114 for CenGEN; neuron class not available in CeSTAAN; #44 for Worm-Seq) to the DA neurons (*Bhatla, 2009*), potentially modulating backward locomotion as part of the chemotaxis response. We also identified pharyngeal neurons I3 (rank #4 for CenGEN; data not available in

CeSTAAN nor Worm-Seq) and I6 (rank #5; neuron class not available in CeSTAAN nor Worm-Seq), which have not previously been implicated in learning. *Figure 4D* provides a summary for the neural circuits implicated from these analyses, where neuron classes are highly connected to each other. Investigating the role of specific genes within these circuits opens new avenues for future research into gustatory learning.

## Validating the requirement of learning proteome components in salt associative learning through single gene studies

Our initial analysis of learning proteome data indicates that there are multiple hits present in biological pathways important for neuron function, and that are potentially relevant to learning and memory formation. To test this directly, we performed salt associative learning experiments on selected learning proteome hits (*Figures 4–6*, *Figure 6—figure supplements 1 and 2*). We used the following general rules to interpret our data: if the average chemotaxis indices (CIs) for 'trained' worms were *higher* in a particular strain compared with wild-type, this strain was considered learning-defective, as this reflects a reduced magnitude of the expected behaviour change (an increased preference for low salt demonstrated by CIs closer to 0 or negative CIs). If the average CI for 'trained' worms was *lower* in a strain compared with wild-type, then this strain was considered to display 'better' learning, as the lower CI reflects an increased magnitude of the expected behaviour change. In general, we observed no significant difference in CIs between naïve groups for all genotypes, reflecting no gross locomotor or chemotaxis defects in the strains tested (*Figures 4–6*, *Figure 6—figure supplements 1–2*).

We tested 26 candidates in total for this study. Although this represents a small subset of the 706 proteins identified in the learning proteome, several proteins in the full list are unsuitable for functional testing due to key constraints: (1) having essential roles, with corresponding single-gene mutants being lethal; (2) involvement in neurodevelopment rather than mature neuronal function; and (3) being required for locomotion, with severe locomotion defects precluding assessment using chemotaxis assays.

Candidates tested were classified as either strong (detected in biological replicates ≥3) or weak (replicates <3) based on the number of mass spectrometry replicates in which they were uniquely identified in TurboID-trained *C. elegans* (shown in brackets). We determined these numbers by considering both assigned and unassigned protein lists, which contained mostly neuron-expressed proteins (*Table 1*) including known learning regulators (*Figure 2* and *Supplementary file 1H*). The list of 26 candidates for further testing includes both weak and strong hits. In addition, although candidates tested were mostly detected in more replicates of trained versus control groups, we also assayed seven candidates for which this was not the case. *Table 2* summarises the potential learning regulators explored in this study, including strong/weak classifications and replicate numbers between experimental groups.

We first tested the regulatory subunit of PKA, *kin-2* (1 replicate), since it is a known regulator of memory and was detected as a weak candidate by TurboID. Adenylyl cyclase is a key signalling effector for Gα$_s$ and Gα$_i$ proteins and regulates levels of the secondary messenger cyclic AMP (cAMP) within the cell. cAMP binding to PKA regulates its activity, and therefore its downstream effects (*Sassone-Corsi, 2012*). We tested worms with the *ce179* mutant allele in *kin-2*, in which a conserved residue in the inhibitory domain (which normally functions to keep PKA turned off in the absence of cAMP) is mutated to cause an R92C amino acid change – this results in increased PKA activity (*Schade et al., 2005*). *kin-2* has previously been shown to be required for intermediate-term memory in *C. elegans* (*Stein and Murphy, 2014*), with cAMP/PKA signalling previously shown to be involved in memory in multiple systems (*Kandel, 2012*). We found that these *kin-2* mutant animals showed enhanced learning compared with wild-type (i.e. Non-Tg worms; *Figure 5A*). We next re-expressed KIN-2(R92C) in wild-type worms using a pan-neuronal promoter, and these worms showed a similar phenotype to *kin-2(ce179)* worms, with enhanced learning compared with non-transgenic siblings (*Figure 5B*). These data suggest that increased PKA activity in the nervous system drives salt associative learning.

We next assessed two strong candidates not previously assessed for their role in learning: putative arginine kinase F46H5.3 (four replicates) or armadillo-domain containing protein C30G12.6 (three replicates). Unlike *kin-2(ce179)* worms, neither single gene mutant obtained from the *Caenorhabditis* Genetics Center had been backcrossed. We backcrossed mutant strains four times to N2 and tested both non-backcrossed and backcrossed versions. An improved learning phenotype was displayed by

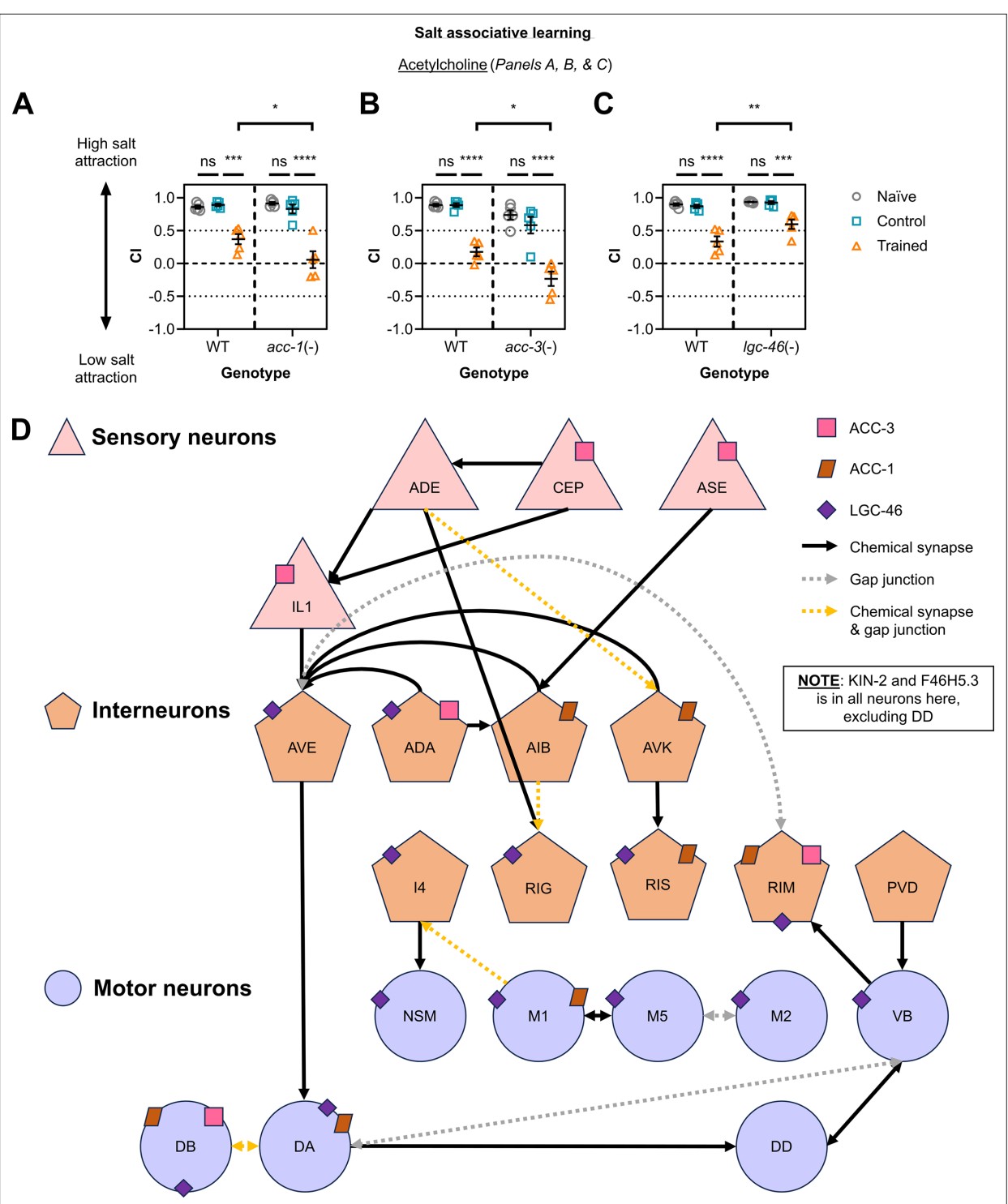

**Figure 4.** Utilising positive candidates involved in cholinergic signalling to illustrate a putative neural circuit containing neuron classes represented by the learning proteome. (**A, B, C**) Chemotaxis assay data for *C. elegans* with mutations targeting cholinergic signalling components *acc-1*, *acc-3*, or *lgc-46*, respectively (n=5). Each data point represents a chemotaxis index (CI) value from one biological replicate (**n**), with three technical replicates per biological replicate (23–346 animals assayed per technical replicate). Error bars = mean ± SEM. Two-way ANOVA and Tukey's multiple comparisons tests were performed to analyse this data (****≤0.0001; ***≤0.001; **≤0.01; *≤0.05; ns = non-significant). (**D**) Neuron classes represented by the learning proteome were identified using the gene enrichment tool from WormBase (***Angeles-Albores et al., 2016***) and the CeNGEN database (threshold = 2; ***Taylor et al., 2021***). Neurons are represented by pink triangles (sensory), orange pentagons (interneurons), and purple circles (motor neurons). Chemical synapse (black arrows) and gap junction (dotted arrows: grey for gap junctions only or yellow for synapses and gap junctions) information is

*Figure 4 continued*

provided using the software WormWeb (*Bhatla, 2009*). Learning regulators validated in this study are also represented: ACC-1 (brown rectangles), ACC-3 (pink squares), and LGC-46 (purple diamonds) are annotated above based on single neuron expression profiles from CeNGEN (*Taylor et al., 2021*). Notably, KIN-2 and F46H5.3, discussed in detail below, are expressed in all neurons shown except for DD.

The online version of this article includes the following figure supplement(s) for figure 4:

**Figure supplement 1.** Analysis of subcellular localisations of assigned protein hits in the learning proteome.

both non-backcrossed and backcrossed *F46H5.3*(-) worms (*Figure 6A and B*). In contrast, we found that non-backcrossed *C30G12.6*(-) animals displayed an enhanced learning phenotype, whereas backcrossed *C30G12.6*(-) mutants behaved like wild-type (*Figure 6C and D*). This suggests that the non-backcrossed *C30G12.6*(-) strain contains a background mutation that impacts learning capacity, a potential avenue for future work.

F46H5.3 is a homolog for creatine kinase B (cytoplasmic) and mitochondrial creatine kinase 1B, but is considered an arginine kinase since *C. elegans* use arginine instead of creatine. Both creatine kinases are essential for energy metabolism via ATP modulation. Given notable representation of metabolic pathways from the learning proteome identified here (e.g. *Figure 3*), this provided additional rationale for testing F46H5.3. Moreover, the depletion of creatine kinase B reportedly increases the latency needed for memory encoding of a spatial learning task in mice (*Jost et al., 2002*). Since F46H5.3 is expressed in most neurons in the worm (*Taylor et al., 2021*)**,** this putative arginine kinase may affect learning capacity through modulating ATP levels in neurons.

We also tested genetic mutants for candidates involved in neurotransmission, a key function for all neurons and a requirement for learning (reviewed in *Myhrer, 2003* and *Rahmani and Chew, 2021*). Two strong candidates, cationic acetylcholine-gated calcium ion channel ACR-2 (four replicates) and cholinergic receptor interactor ELP-1 (three replicates), were not observed to regulate

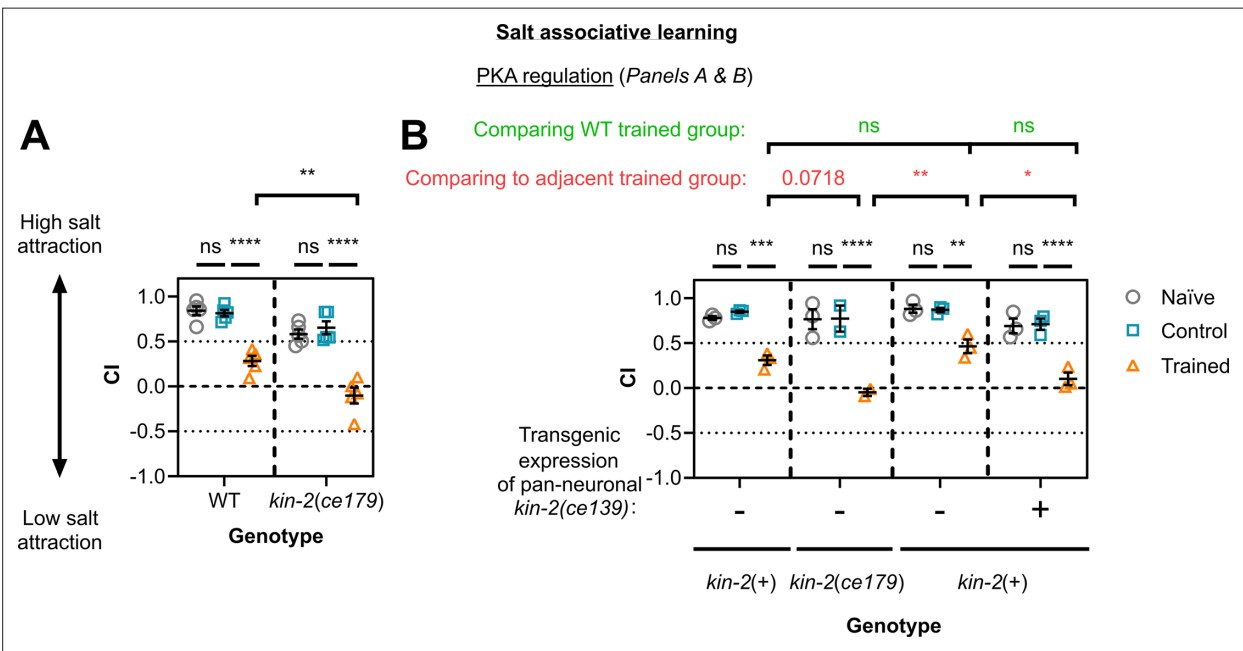

**Figure 5.** *C. elegans* PKA regulatory subunit KIN-2 acts in neurons to regulate salt associative learning. Salt chemotaxis behaviour was measured in the form of chemotaxis indices (CI) for naive/untrained worms (grey circles), high-salt control (blue squares), and trained worms (orange triangles). This was done for (A and B) wild-type (WT) animals, (A and B) *kin-2(ce179)* mutants, and (B) transgenic worms with a WT background engineered to overexpress KIN-2 from the *ce179* allele in all neurons (10–60% transgenic worms per technical replicate, both non-transgenic (-) and transgenic (+) siblings are plotted above). Each data point represents one biological replicate where (**A**) n=5 and (**B**) n=3 (one biological replicate was excluded from high-salt control and trained *kin-2(ce179)* groups due to insufficient sample size). (**A**) 32–487 worms and (**B**) 5–184 worms per technical replicate. Error bars = mean ± SEM. Annotations above graphs represent P-values from Two-way ANOVA and Tukey's multiple comparison tests (****≤0.0001; ***≤0.001; **≤0.01; *≤0.05; ns = non-significant). (**B**) Statistical comparisons between WT trained and siblings in transgenic lines are in red (top row), between adjacent trained groups are in green (middle row), and between groups within each line in black (bottom row).

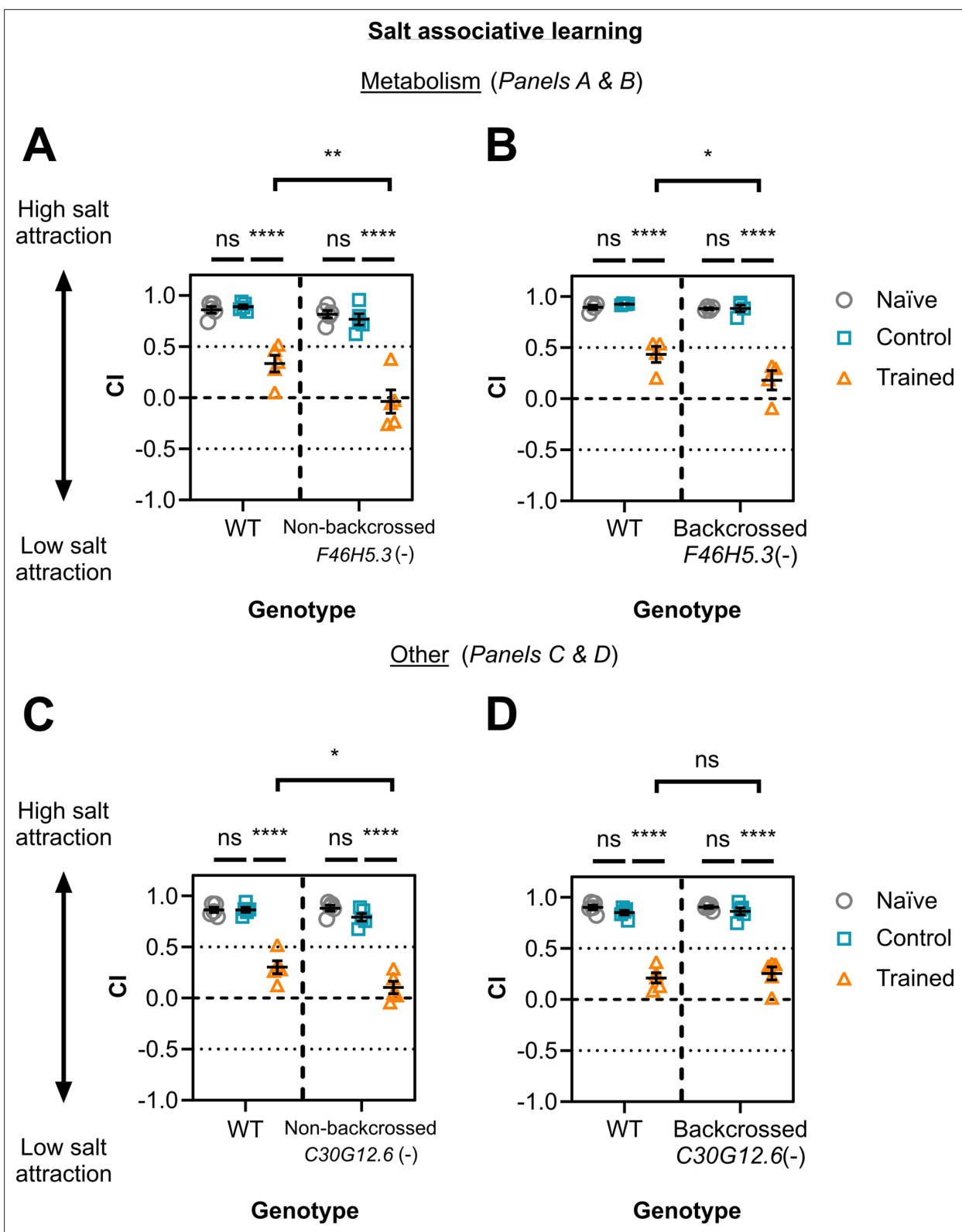

**Figure 6.** Salt associative learning is dependent on arginine kinase F46H5.3 and not armadillo-domain containing protein C30G12.6. Chemotaxis indices (CI) are shown for wild-type/WT animals versus mutants for (**A**) F46H5.3 (non-backcrossed with WT, n=5), (**B**) F46H5.3 backcrossed with WT (n=4), (**C**) C30G12.6 (non-backcrossed with WT, n=5), and (**D**) C30G12.6 backcrossed with WT (n=5). These animals were assessed for salt associative learning by preparing three groups for each line: naïve/untrained (grey circles), high-salt control (blue squares), and trained (orange triangles; 27–395 worms

*Figure 6 continued on next page*

*Figure 6 continued*

per technical replicate). Each data point is for one biological replicate each comprising three technical replicates. Error bars = mean ± SEM. Statistical analyses were done by Two-way ANOVA and Tukey's multiple comparison test (****≤0.0001; **≤0.01; *≤0.05; ns = non-significant).

The online version of this article includes the following figure supplement(s) for figure 6:

**Figure supplement 1.** Behavioural testing for strong candidates identified by TurboID-based mass spectrometry experiments.

**Figure supplement 2.** Behavioural testing for weak candidates in the learning proteome.

**Figure supplement 3.** Salt aversive learning is modulated by arginine kinase F46H5.3.

**Table 2.** Summary of candidates assessed for their effect in learning.

The number (#) of biological replicates (total n=5) in which each candidate was detected as an assigned hit (by the MASCOT software) or in assigned + unassigned hits (identified by bulk BLAST search) is provided under '# Biological replicates in TurboID trained' and ' # Biological replicates in TurboID high-salt control' columns. These values exclude proteins from non-transgenic trained and non-transgenic high-salt control groups, respectively. Orange highlights indicate candidates detected in more replicates in the TurboID-trained group. Candidates are also defined as 'weak' or 'strong' based on the frequency of detection across biological replicates.

| Candidates tested | # Biological replicates in *TurboID trained* (assigned hits) | # Biological replicates in *TurboID high-salt control* (assigned hits) | # Biological replicates in *Turbol D trained* (assigned + unassigned hits) | Classification for candidate |
|---|---|---|---|---|
| IFT-139 | 4 | 1 | 5 | Strong |
| ACR-2 | 1 | 0 | 4 | Strong |
| F46H5.3 | 3 | 2 | 4 | Strong |
| SAEG-1 | 2 | 0 | 4 | Strong |
| UEV-3 | 4 | 1 | 4 | Strong |
| AEX-3 | 0 | 0 | 3 | Strong |
| C30G12.6 | 0 | 0 | 3 | Strong |
| ELO-6 | 3 | 0 | 3 | Strong |
| ELP-1 | 2 | 1 | 3 | Strong |
| FSN-1 | 0 | 0 | 3 | Strong |
| GAP-2 | 2 | 0 | 3 | Strong |
| RIG-4 | 0 | 1 | 3 | Strong |
| TAG-52 | 1 | 0 | 3 | Strong |
| TAP-1 | 2 | 0 | 3 | Strong |
| VER-3 | 3 | 0 | 3 | Strong |
| ACC-3 | 1 | 0 | 2 | Weak |
| DLK-1 | 1 | 0 | 2 | Weak |
| GBB-2 | 2 | 0 | 2 | Weak |
| GPA-2 | 2 | 0 | 2 | Weak |
| RHO-1 | 2 | 1 | 2 | Weak |
| ACC-1 | 1 | 1 | 1 | Weak |
| GAP-1 | 1 | 0 | 1 | Weak |
| GLR-1 | 0 | 1 | 1 | Weak |
| KIN-2 | 1 | 0 | 1 | Weak |
| LGC-46 | 1 | 1 | 1 | Weak |
| MACO-1 | 1 | 0 | 1 | Weak |

learning (*Figure 6—figure supplement 1A and B*). Interestingly, several acetylcholine receptors that were weak candidates showed differences in learning compared with wild-type controls. Those tested include ACC-1 (one replicate), ACC-3 (2 replicates), and LGC-46 (1 replicate) in the subfamily of anionic 'ACC' acetylcholine-gated ligand-gated ion channels (*Morud et al., 2021*). Interestingly, *lgc-46* mutants demonstrate a reduced capacity for learning compared with wild-type, whereas *acc-1* and *acc-3* mutants appear to display better learning (*Figure 4A, B and C*). Although these ACC receptors are activated by acetylcholine (*Park et al., 2000*; *Putrenko et al., 2005*), or have been reported to possess the same protein domain/s as known acetylcholine receptors (*Takayanagi-Kiya et al., 2016*), they vary substantially in expression pattern in the nervous system (*Taylor et al., 2021*). We postulate that acetylcholine signalling in specific neurons may therefore contribute to learning in different directions, an interesting avenue for future research. We also tested other genes involved in neurotransmission, such as *gbb-2* (for GABAergic signalling, 2 replicates) and *glr-1* (for glutamatergic signalling, 1 replicate), neither showed a detectable change in learning capacity compared with wild-type (*Figure 6—figure supplement 2A and B*). *maco-1* (1 replicate) encodes a macoilin family protein that functions broadly in neurotransmission (*Arellano-Carbajal et al., 2011*), and has previously been shown to regulate memory of olfactory adaptation in worms (*Kitazono et al., 2017*). The *maco-1* mutant tested in our study (*nj21*) did not show a locomotor defect but also did not show any obvious learning phenotypes (*Figure 6—figure supplement 2C*). Our data, therefore, indicate that specific components of cholinergic signalling are required for salt associative learning.

We tested several components of GPCR signalling that were identified as learning proteome hits, including *gap-1* (one replicate), *gap-2* (three replicates), *gpa-2* (two replicates), and *rho-1* (two replicates; *Figure 6—figure supplements 1C, D and 2E, F*), but found that worms with single mutations for these genes, when compared to wild-type controls, did not show statistically significant differences in learning capacity. We postulate that although GPCR signalling is broadly important for learning (reviewed in *Jong et al., 2018* and *Rahmani and Chew, 2021*), there may be high levels of redundancy built within this pathway such that single pathway components can be compensated for by other functionally similar genes.

Several components of MAPK signalling have been shown to be involved in different forms of learning (reviewed in *Peng et al., 2010*; *Ryu and Lee, 2016*), including NSY-1/MAPKKK, MEK-2/MAPKK, and MPK-1/ERK that were identified as part of the learning proteome in our study (*Ohno et al., 2014*). We tested the dual leucine zipper MAPKKK-encoding gene *dlk-1* (two replicates), a mutant of this gene showed no difference in learning capacity compared to wild-type (*Figure 6—figure supplement 2G*). The E2 ubiquitin-conjugating enzyme variant UEV-3 (four replicates) has been shown to be a member of the DLK-1 pathway and a potential interactor of p38/MAPK PMK-3 (*Trujillo et al., 2010*); trained phenotypes between wild-type and *uev-3* mutants were not statistically significant (*Figure 6—figure supplement 1D*). We also tested worms with a mutation in *fsn-1* (three replicates), proposed to attenuate synapse growth in a DLK-1-dependent manner (*Hung et al., 2013*). These animals displayed learning capacity similar to wild-type animals (*Figure 6—figure supplement 1E*). In summary, these data indicate that mutating single components of the MAPK pathway does not generally perturb salt associative learning.

Finally, we assessed additional hits that do not fit in the pathways above but were considered strong candidates. These include neuronal adhesion/IGCAM gene *rig-4* (three replicates) or putative guanyl-nucleotide exchange factor genes (*aex-3* or *tag-52*, both three replicates). Aversive associative learning in worms and mice both relies on IGCAM gene *ncam-1/NCAM1* (*Cremer et al., 1994*; *Vukojevic et al., 2020*; *Doyle et al., 1992*). Additionally, guanine nucleotide exchange factors 'ArhGEF4' and 'RapGEF2' in mice (*Jiang et al., 2024*; *Yoo et al., 2020*), as well as *unc-73* in *C. elegans* (*Arey et al., 2018*), have been linked to learning and memory previously. We found that *aex-3*(-), *rig-4*(-), and *tag-52*(-) single mutants did not show significant differences in salt associative learning compared to wild-type controls (*Figure 6—figure supplement 1F, G and H*). We also tested *elo-6* (three replicates), which encodes a long chain fatty acid elongase that potentially functions together with another elongase encoded by *elo-5*, although only *elo-6* is expressed in neurons (*Kniazeva et al., 2004*). Fatty acid composition has previously been demonstrated to be important for learning and memory (*Wallis et al., 2021*; *Pershina et al., 2022*; *Akefe et al., 2024*). Our data showed no significant learning defect or improvement in *elo-6* mutant animals (*Figure 6—figure supplement 1I*), although, as mentioned, its role may be masked by functional redundancy.

Other strong candidates tested that did not show a learning phenotype include: (1) *ift-139* (5 replicates), a ciliogenesis gene that was explored since hippocampal cilia structures have been seen to be important for memory in mice (**Niwa, 2016**; **Jovasevic et al., 2021**; **Berbari et al., 2014**). (2) *tap-1* (three replicates), which encodes an ortholog for TGF-β activated kinase 1 (**Meneghini et al., 1999**), (3) SAEG-1 (four replicates), a suppressor for protein kinase G (PKG) ortholog EGL-4 activity, which has been implicated in behavioural changes induced by odour-sensory fatigue **L'Etoile et al., 2002**; cGMP-dependent kinase PKG also promotes long-term memory in rodents (**Ota et al., 2008**; **Paul et al., 2008**), and (4) VER-3 (three replicates), which encodes a predicted vascular endothelial growth factor (VEGF) receptor-like protein (**Popovici et al., 2002**). VEGF/VEGFR was reported to be upregulated in rats following spatial learning (**Cao et al., 2004**). These data are shown in *Figure 6—figure supplement 1*.

There are several potential reasons why many mutants tested did not display a learning phenotype. Firstly, as mentioned above, effects may be masked by redundant or compensatory pathways. For example, IGCAM genes in the worm have been reported to act redundantly in axon navigation, including *rig-4* (**Schwarz et al., 2009**). It is also possible that these mutations do not fully knock out protein function. We generally assessed animals with deletion mutations predicted to disrupt protein function, but we did not confirm this through qualitative or quantitative means. Additionally, many mutants used here were not backcrossed as it was beyond the scope of this study, so these lines may have background mutations masking learning phenotypes of the mutations of interest. This was seen for *C30G12.6*(-) animals in this study, where an enhanced learning phenotype in non-backcrossed worms was lost after backcrossing (*Figure 6*). Separately, although memory encoding (learning) and retention are interlinked biological processes, they are molecularly distinct (**Arey et al., 2018**; **Kauffman et al., 2010**; **Watteyne et al., 2020**; **Rahmani and Chew, 2021**). Typically, learning is assayed for *C. elegans* immediately after training, whereas memory retention is assessed after a post-training rest period – this assessment usually relies on a behavioural change representing learning/memory formation. It is possible that some candidates detected in neurons during learning may correspond to those important for memory retention and not encoding, so it may be worth utilising this proteomic dataset as a resource to explore this concept in future. Finally, the population-wide chemotaxis assays we perform here to validate candidates may not be sensitive enough to capture subtle potential behavioural differences caused by these mutations. Pirouette and weathervane behaviours in *C. elegans* change based on previously experienced salt concentrations in the presence of food (**Kunitomo et al., 2013**). These behaviours can be measured through more in-depth investigation of locomotor behaviour through live tracking and analysis, providing a more sensitive measure for learning responses compared to the chemotaxis assays used here. These factors are important considerations for future experiments utilising the learning proteome as rationale to assess novel mechanisms in learning.

## Arginine kinase F46H5.3 regulates both appetitive and aversive gustatory learning

Next, we explored whether the learning regulators identified from our learning proteome functioned more broadly in other types of learning. To do this, we assayed salt aversive learning capacity (training with aversive cue i.e. starvation) for candidates seen to affect salt associative learning (training with an appetitive cue i.e. presence of food): PKA regulatory protein KIN-2 (*Figure 5*), arginine kinase F46H5.3 (*Figure 6*), and acetylcholine receptor subunits ACC-1, ACC3, and LGC-46 (*Figure 4*). In the salt associative learning assay used thus far, 'trained' worms are exposed to a pairing of no salt + food ('control' worms with high salt +food), whereas in salt aversive learning, 'conditioned' worms are exposed to high salt + no food (and control 'mock-conditioned' worms with no salt + no food). In the salt aversive learning assay, 'conditioned' worms therefore learn to avoid high salt (as it is associated with starvation, a strongly negative cue), whereas 'mock-conditioned' worms and naïve worms retain a preference for high salt (**Nagashima et al., 2019**; **Hiroki et al., 2022**). We found that only *F46H5.3*(-) mutant worms showed a significant change in learning capacity for salt aversive learning compared with wild-type (*Figure 6—figure supplement 3*). Specifically, *F46H5.3*(-) mutants displayed a larger decrease in CI in trained animals compared to wild-type trained worms, demonstrating a potential learning improvement (*Figure 6—figure supplement 3A*). *F46H5.3*(-) mutant phenotype also showed enhanced learning for the salt associative learning paradigm (*Figure 6A and B*). Although

*kin-2(ce179)* mutants were not shown to impact salt aversive learning, they have been reported previously to display impaired intermediate-term memory (but intact learning and short-term memory) for butanone appetitive learning (*Stein and Murphy, 2014*). These findings therefore suggest (1) a generalised effect for F46H5.3 in gustatory learning paradigms involving salt, (2) a specific role for KIN-2 in appetitive learning paradigms, and (3) a unique effect for ACC receptors ACC-1/2 and LGC-46 in salt associative learning only.

## Using TurboID to predict potential molecular and cellular pathways for learning

Our TurboID approach provides a unique benefit as a systems-based tool, in that it can be used to map individual candidates onto broader molecular networks. F46H5.3 is mostly uncharacterised beyond its predicted homology to creatine kinase B and mitochondrial creatine kinase 1B. As its role within learning pathways is unknown, we tested for protein-protein interactions between F46H5.3 and other candidates in the learning proteome using the software STRING (version 12.0; *Szklarczyk et al., 2023*), aiming to infer its potential function within a molecular network. Mitochondrial creatine kinases regulate phosphocreatine synthesis using ATP, which requires calcium influx into mitochondria to induce ATP synthesis (reviewed in *Schlattner et al., 2001*; *Schlattner et al., 2006*). Voltage-dependent anion channel VDAC-1 (identified in three replicates of learning proteome data) plays a critical role in calcium homeostasis in *C. elegans* mitochondria (*Shoshan-barmatz et al., 2017*) and is predicted interactor of F46H5.3. Moreover, calcium influx into the mitochondria is regulated by calcium/calmodulin kinase II (*Nguyen et al., 2018*), an established and highly conserved regulator of learning (reviewed in *Ataei et al., 2015*; *Zalcman et al., 2018*). The sole calcium/calmodulin kinase II in *C. elegans* (UNC-43, two replicates) is predicted to interact with VDAC-1. Our learning proteome also includes proteins involved in calcium/calmodulin complex formation (e.g. calmodulin/CMD-1 in four replicates and cyclic nucleotide–gated ion channel TAX-4 in one replicate; *Karabinos et al., 2003*; *Komatsu et al., 1999*). Calcium/calmodulin complexes can also modulate cAMP levels (reviewed in *Sharma and Kalra, 1994*), which influences PKA/KIN-1 activity regulation by KIN-2 and A-kinase anchoring protein AKAP-1 (two replicates; reviewed in *Sadeghian et al., 2022*). Therefore, cytoplasmic calcium homeostasis is one potential pathway through which both KIN-2 and F46H5.3, validated in our study as learning regulators, modulate learning and memory (*Figure 7*).

Finally, we can combine the protein network analysis with analysis of neuron representation within the learning proteome, as described above and shown in *Figure 4D*. There are two neuron classes that express all five candidates shown here to affect salt associative learning (KIN-2, F46H5.3, ACC-1, ACC-3, LGC-46): RIM interneurons and DB motor neurons (*Taylor et al., 2021*; threshold = 2). These neurons are involved in reversals and forward locomotion, respectively (*Guo et al., 2009*; *Chalfie et al., 1985*). It is possible that these learning regulators influence experience-based behaviour through modulating the function of these neurons to alter chemotaxis responses in the presence of gustatory cues.

KIN-2 and F46H5.3 share the same expression pattern in many neuron classes, whereas neurons expressing the three ACC receptors are more diverse (*Taylor et al., 2021*; *Figure 4D*). ACC-3 is expressed in salt-sensing ASE, mechanosensory and dopaminergic CEP neurons, polymodal sensory neuron IL1, and two interneuron classes (ADA and RIM) (*Taylor et al., 2021*). ADA's function is not well-characterised, but it is predicted to be involved in chemosensation (*Sohn et al., 2011*). In contrast, ACC-1 and LGC-46 are expressed in several interneurons and motor neurons including those implicated in gustatory or olfactory learning paradigms (AIB, AVK, NSM, RIG, and RIS; *Beets et al., 2012*; *Fadda et al., 2020*; *Wang et al., 2025*; *Zhou et al., 2023*; *Sato et al., 2021*) and important for backward or forward locomotion (AVE, DA, DB, and VB; *Chalfie et al., 1985*). There are also highly represented neuron classes which are not as well defined (ADA, I4, M1, M2, and M5), which may present interesting directions for future research. Cholinergic signalling may therefore regulate gustatory learning through integration of sensory signals as well as direct modulation of the motor circuit. In contrast, KIN-2 and F46H5.3 may play more general functions within the nervous system, such as through modulating calcium homeostasis (*Figure 7*). As mentioned above, *Figure 4D* utilises the CeNGEN database generated from L4 animals, so we cross-referenced this expression data with transcriptome studies using young adult hermaphrodites (as in this study). Our five candidates of interest (ACC-1/3, F46H5.3, KIN-2, and LGC-46) were reported in the same neuron classes

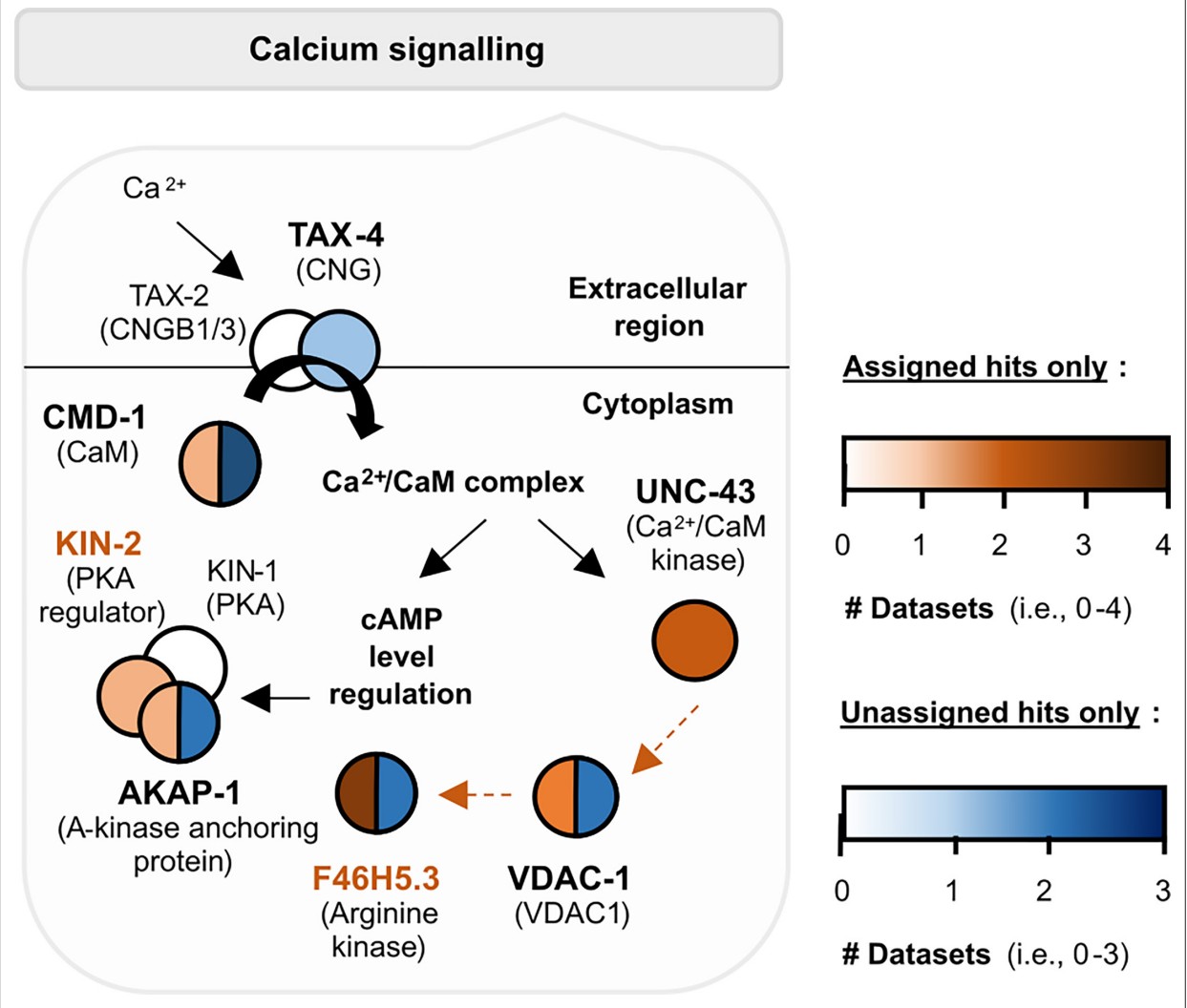

**Figure 7.** Learning regulators KIN-2 and F46H5.3 may modulate learning through calcium signalling pathways. Pathway components present within the learning proteome are shown with protein names in bold. Darker colours mean the protein has been detected in more than one biological replicate (see legend). Orange and/or blue circles represent candidates that are 'assigned hits' and/or 'unassigned hits', respectively. Orange dotted arrows denote protein-protein interactions predicted by STRING (version 12.0), whereas black arrows are based on known interactions.

as in CeNGEN, further suggesting that these neurons may be important for memory encoding in the worm (*Ghaddar et al., 2023*; *Kaletsky et al., 2016*; *Roux et al., 2023*; *Smith et al., 2024*; *St Ange et al., 2024*). Overall, our TurboID dataset offers a valuable foundation for future investigations into individual proteins involved in learning and provides a resource for systemic analyses at the level of tissues, neuron types, subcellular localisations, and molecular networks.

## Discussion

Our study demonstrates the effectiveness of using the protein labelling technique TurboID to explore *C. elegans* learning and memory. The expression of TurboID in the whole nervous system of the worm, and addition of biotin only during the training step of salt associative learning, allowed us to label proteins in neurons during this critical stage of memory encoding. We identified these proteins by mass spectrometry and revealed a putative 'learning proteome' including known learning regulators (*Figure 2*, *Supplementary file 1H*). Moreover, we identified five novel regulators of appetitive gustatory learning, namely three acetylcholine receptors (ACC-1, ACC-3, and LGC-46), PKA regulatory subunit KIN-2, and the arginine kinase F46H5.3 (*Figures 4–6*). Finally, F46H5.3 was observed to

modulate an aversive gustatory learning paradigm in *Figure 6—figure supplement 3*. These findings highlight that proximity labelling can be used in *C. elegans* to elucidate novel learning regulators, which may function across learning paradigms characterised by different modalities or valences.

Learning and memory are key functions of the nervous system and are critical for survival. Forms of associative learning have been studied in invertebrate and vertebrate animals for decades, revealing many important insights on the behavioural, neuroanatomical, and molecular requirements for learning and memory formation (reviewed in *Hawkins and Byrne, 2015*, *Jong et al., 2018*, *Kandel, 2012*, *Matsumoto et al., 2018*, *Peng et al., 2010*, *Peng et al., 2011*, *Rahmani and Chew, 2021*, *Ryu and Lee, 2016*). However, many studies focus on single genes of interest and are therefore unable to reveal the entire network of molecular players that drive (or inhibit) learning. We used a complementary approach to take a snapshot of the proteins present in the brain during learning, using the protein-labelling tool TurboID. Previously, microarray analysis had been done to charac-terise transcriptomic changes during long-term memory formation in the *C. elegans* nervous system (*Lakhina et al., 2015*). While this provides useful insights into the mechanisms of memory forma-tion, it does not capture proteomic information, which may differ from RNA levels. Separately, *Hiroki and Iino, 2022* applied TurboID to map protein-protein interactions of PKC/PKC-1 in untrained *C. elegans*, providing insight into how PKC-1 affects gustatory appetitive learning. Our work builds on this by directly comparing trained and high-salt control conditions, offering new insights into the proteomic landscape of learning. Our strategy identified several components of molecular pathways previously shown to be generally required for learning, including neurotransmitter signalling, MAPK signalling, insulin signalling, synaptic vesicle exocytosis, and GPCR signalling (*Figure 2*, *Figure 2—figure supplement 1*, and *Supplementary file 1H*). We identified regulators of learning that may not have been obvious choices for a candidate screen and that may have had phenotypes too subtle to be highlighted through a random mutagenesis screen. We therefore conclude that our approach is a useful and scalable method that can be used in multiple systems to delineate the molecular require-ments for different forms of learning.

There are several interesting unanswered questions: Firstly, why did some candidates only seem to affect gustatory appetitive learning, as opposed to showing effects in both gustatory appetitive and aversive paradigms? Mutants of ACC-1, ACC-3, LGC-46, and KIN-2 show significantly different learning capacities for salt associative learning compared with wild-type (*Figures 4 and 5*) but did not show differences when tested for salt aversive learning (*Figure 6—figure supplement 3*). In contrast, only *F46H5.3*(-) mutants showed a significant learning difference in both salt associative learning and salt aversive learning (*Figure 6*, *Figure 6—figure supplement 3A*). One possibility is that our method for selecting hits from our mass spectrometry data for downstream validation introduced an unin-tended bias: to identify the learning proteome, we subtracted proteins in the list for 'high-salt control' worms from the protein list for 'trained' worms, for each biological replicate. The main difference between high-salt control and trained worms was whether they were exposed to a pairing of 'salt + food' (control) or 'no salt + food' (trained). It is possible that proteins present in trained worms, but not high-salt control worms, during conditioning are those that are strongly regulated by changes in salt concentration (and therefore impact mainly appetitive gustatory learning with 'no salt', versus salt aversive learning induced in the presence of salt). Indeed, our 'simple subtraction' approach may be an overly conservative method for selecting learning regulators, as it is likely that many neuronal proteins generally important for learning (i.e. across multiple learning paradigms) are present in both groups, but at higher (or lower) levels in trained worms.

Secondly, why do the different acetylcholine receptors that we identified in our study impact learning differently? We showed that loss-of-function mutants of ACC-1 and ACC-3 show improved learning, whereas *lgc-46*(-) mutants displayed a learning defect compared with wild-type (*Figure 4*). One study suggested that ACC-1 and ACC-3 may function together: ACC-3 homomers do not respond robustly to acetylcholine, but ACC-1 and ACC-3 can form a functional heteromer, albeit with lower sensitivity to acetylcholine than ACC-1 homomers (*Putrenko et al., 2005*). ACC-1 and ACC-3 functioning as heteromers may be why these proteins impact learning in the same direction. On the other hand, LGC-46 is also a member of the acetylcholine-gated chloride channel (ACC) family but impacts salt associative learning in the opposite direction to ACC-1 and ACC-3. LGC-46 (in 81 neurons) has a much broader expression pattern than ACC-1 (in 32 neurons) or ACC-3 (in 14 neurons) and is expressed in many more interneurons and motor neurons (*Taylor et al., 2021*). It is therefore possible that some

of the LGC-46-expressing neurons function to regulate learning in a different manner to ACC-1- or ACC-3-expressing cells.

Our approach demonstrates a powerful method to uncover regulatory networks for a variety of behaviours; however, one factor we aim to improve in the future is the amount of background 'noise' observed in the learning proteome. While the five biological replicates of learning proteome data did reveal genes and molecular pathways implicated in learning and memory, there was also potential background. For example, when assessing the overlap between all proteins (i.e. both assigned and unassigned proteins *Figure 1—figure supplement 4*), a large proportion of genes identified were categorised as ribosomal (4–12%), mitochondrial (8–12%), or involved in reproduction (16–19%; STRING database, accessed Feb 2024). Mass spectrometers detect peptides by abundance (*Bakalarski et al., 2008*), so many of these highly abundant proteins may have preferentially been detected over less abundant neuronal proteins. There are several ways to reduce background and improve the signal-to-noise ratio: (1) using an integrated transgenic line that expresses TurboID with 100% transmission, as the line used in our study had a 70–80% transmission rate, (2) using a cell-sorting strategy (e.g. by flow cytometry) to isolate a tissue of interest *Beets et al., 2020*; this is particularly useful if the tissue of interest is only a low proportion of the worm biomass (e.g. neurons are 1% *Froehlich et al., 2021*), as proteins from larger tissues, such as the gut and germline, may interfere with detection, and (3) removing highly abundant background proteins during mass spectrometry sample preparation, such as mitochondrial carboxylases that are endogenously biotinylated, as in *Artan et al., 2022*.

There are also inherent limitations to using a qualitative approach. While our approach includes weak candidates and does not include a statistical framework for comparing protein abundance between experimental groups, this flexibility allows for the identification of potentially novel regulators that might otherwise be overlooked in more stringent analyses. Notably, we did observe relationships between weak candidates and learning. For example, ACC-1, which modulates salt associative learning in *C. elegans*, was detected in one replicate of mass spectrometry as a potential learning regulator (*Figure 4A*). To address the lack of quantitative comparison, we categorised each candidate with their occurrence per replicate of mass spectrometry data for TurboID trained versus high-salt control datasets (summary of this data for candidates tested shown in *Table 2*). In addition, the raw mass spectrometry data is provided for each biological replicate and experimental group via an open-access server (see 'Data Availability'), enabling transparency and further analysis by the research community. Future studies could benefit from implementing a quantitative approach to directly measure protein abundance differences between trained and control groups. While integrating such approaches with TurboID is challenging due to the requirement for biotinylated protein enrichment, overcoming these limitations, or using an alternative proteomic strategy, could uncover additional learning regulators.

Finally, while the learning regulators identified in this study support the validity of our proteomic approach, further functional validation is important. Testing for rescue of learning phenotypes in transgenic lines re-expressing learning regulators pan-neuronally, endogenously, or in single neurons would provide valuable insight into their functions within the nervous system. Our attempts to generate such rescue lines using standard microinjection techniques encountered several technical challenges, including difficulties with low transmission rate, potentially due to plasmid toxicity, and culturing issues potentially caused by transgene-dependent reproductive defects. To overcome these challenges, future work may utilise single-copy integration methods to reduce transgene dosage or use tissue- or cell-specific RNA interference to achieve targeted knockdown. These approaches could provide more precise insights into the roles of learning regulators within specific neuronal contexts.

In conclusion, we present an effective and scalable approach to identify the network of molecular processes that drive learning and memory formation, using the compact *C. elegans* nervous system. Our data reveal proteins from established biological pathways linked to associative memory, and through which we have identified novel regulators of gustatory associative learning. Future studies using this approach to identify learning regulators in other contexts will advance our understanding of the complex spatiotemporal regulation of learning and memory. This may help to elucidate the principles through which different memory types arise from the combination of specific neuronal signals, individual brain regions/cells, and different sensory modalities, relevant to brains of many sizes.

## Materials and methods

### *C. elegans* strain maintenance

Young adult (day 1) hermaphrodite *C. elegans* were grown using standard conditions on nematode growth medium (NGM) agar in petri dishes at 22 °C for all experiments (*Brenner, 1974*). This was done for at least two generations for salt associative learning assays involving TurboID, as well as assays involving salt aversive learning involving *kin-2(ce179)* mutants. Animals were otherwise cultured at 22 °C for one generation and at 15 °C prior to this. For TurboID-based labelling experiments, worms were cultured for at least two generations with the biotin-auxotrophic strain as their food source (*Branon et al., 2018*). For all other experiments, animals were fed *Escherichia coli* (*E. coli*) strain OP50. *C. elegans* lines used in this paper are listed in *Supplementary file 1A*. Information about plasmids used to generate lines YLC207 and YLC369 is provided as Source Data.

### Biotin treatment

The biotin treatment strategy was adapted from *Artan et al., 2021*. Briefly, a solution of 100 mM biotin (in 250 mM KOH, 5 mM $K_3PO_4$ (pH 6.0), 1 mM $CaCl_2$, and 1 mM $MgSO_4$) was diluted 1:100 in *E. coli* MG1655 *bioB::kan* washed with modified Luria Broth (LB; 25 mM NaCl, 5 mM $K_3PO_4$ (pH 6.0), 1 mM $CaCl_2$, 1 mM $MgSO_4$, 1.0% (w/v) Bacto Tryptone, 0.5% (w/v) yeast extract, 0.05 mg/mL Kanamycin). Biotin-depleted worms were fed *E. coli* MG1655 on NGM agar during a 6 hr conditioning period (see the 'Salt associative learning' section). The bacterial pH was increased by a negligible amount (i.e. 0.1) from the addition of KOH; thus, it is expected it will not significantly impact worm physiology (*Khanna et al., 1997*; *Cong et al., 2020*). At least 3000 worms per group were utilised for downstream proteomic experiments.

### Protein extraction and quantification

The following protocol was adapted from *Liang et al., 2014* and *Artan et al., 2021*: After biotin treatment (end of learning assay), worms were washed twice using 'washing buffer' (50 mM NaCl, 5 mM $K_3PO_4$ (pH 6.0), 1 mM $CaCl_2$, 1 mM $MgSO_4$) and then stored as pellets in a –80 °C freezer. Each pellet was suspended in 200 µL Radioimmunoprecipitation assay (RIPA) buffer containing 2 M urea (prepared as in *Sanchez and Feldman, 2021*), as well as 150 mM NaCl, 50 mM Tris-Cl (pH 8.0), 5 mM EDTA, 10 mM NaF, 2 mM $Na_3VO_4$, 1 mM NaPP, 1% (v/v) Nonidet-P40, 1% (w/v) SDS, 0.5% (w/v) sodium deoxycholate, 0.1% (w/v) β-glycerophosphate, and 1×cOmplete Mini Protease Inhibitor (Merck). Worm pellets were sonicated (10×4 s total sonication time, 2 s 'on' pulse, 3 s 'off' pulse', and 25% amplitude) with the Q125 Sonicator (Q Sonica) in a temperature-controlled room (~2–8°C) and allowed to rest on ice for >20 s between each sonication. All samples were vortexed for ~5 s at room temperature, and then centrifuged (14,000 rcf, 4 °C, 10 min) to separate carcasses/debris from supernatant containing proteins. The supernatant was then used for protein quantification by BCA assay (Thermo Fisher Scientific, #23225).

### SDS-PAGE, protein transfer, and western blotting

20–40 µL protein samples, containing 1×sample buffer (0.25 M Tris (pH 6.8), 10% (v/v) β-mercaptoethanol, 10% (v/v) glycerol, 10% (w/v) SDS, 0.25% (w/v) bromophenol blue), were boiled at 95 °C for 3 min. Proteins were electrophoresed through 8% (v/v) polyacrylamide gels for 40–60 min at 80 V, before the voltage was increased to 100 V for an additional 40–60 min, and then increased to 120 V until completion of electrophoresis (*Jeong et al., 2018*; *Stefanoska et al., 2022*). The standard protocol was utilised for semi-dry protein transfer (constant 25 V, 40–45 min) onto nitrocellulose membranes and western blotting, using the following antibodies/probes (dilution in brackets, all made in 5% (w/v) bovine serum albumin): rabbit anti-α tubulin (1:1000; Abcam, #ab4074), mouse anti-V5 (1:1000; Cell Signalling Technology, #80076), goat anti-rabbit HRP (1:20,000; Cell Signalling Technology, #7074P2), goat anti-mouse HRP (1:20,000; Thermo Fisher Scientific, #G-21040), and streptavidin-HRP (1:5000; Cell Signalling Technology, #3999; *Jeong et al., 2018*; *Stefanoska et al., 2022*). Probed proteins were visualised by chemiluminescence using Clarity Western ECL Substrate (Bio-Rad, #1705060) according to manufacturer's instructions.

## Mass spectrometry

### Sample preparation

Our protocol was adapted from *Artan et al., 2022*, *Sanchez et al., 2021*, and *Prikas et al., 2020*. ~1 mg of protein was desalted using 7 kDa molecular weight cut-off desalting spin column (Thermo Fisher Scientific, #89883) by buffer exchange with RIPA buffer containing a lower SDS content (i.e. 0.1% (w/v) SDS) and no urea (*Artan et al., 2022*). Desalted protein samples were quantified by BCA assay so that ~0.6–1.0 mg of total protein per sample could be used in subsequent pull-down experiments to enrich for biotinylated peptides using Streptavidin Magnetic Beads (NEB, #S1420S) equilibrated with TBS-T (150 mM NaCl, 10 mM Tris (pH 7.4), 0.1% (v/v) Tween-20). Total protein was gently agitated in a tube rotator in 1 mL of total volume (4.5 µL bead:8.0 µg total protein ratio) for 18 hr at 4 °C. Magnetic beads were sequentially washed with the following solutions (number of washes in brackets): TBST (x3), 1 M KCl (1 x), 0.1 M $Na_2CO_3$ (1 x), and PBS (5 x; Therm Fisher Scientific, #10010023; *Sanchez et al., 2021*).

The beads were then incubated in a ThermoMixer (Eppendorf # 5384000063) at 800 rpm, 55 °C for 1 hr in 200 µL per sample of reducing solution containing 4 M urea, 50 mM $NH_4HCO_3$, 5 mM dithiothreitol, and 0.1% (w/v) Protease-Max Surfactant (Promega, #V2071) in 50 mM $NH_4HCO_3$. Alkylation was promoted by adding 4 µL of 0.5 M iodoacetamide to each solution and re-incubating each sample at 800 rpm, 55 °C for 20 min in the dark. Finally, samples were incubated at 800 rpm, 37 °C, for 18 hr with the addition of 162 µL of digesting solution per sample (50 mM $NH_4HCO_3$, 0.1% (w/v) Protease-Max Surfactant (Promega, #V2071) in 50 mM $NH_4HCO_3$, 0.01 µg/µL of Sequencing Grade Modified Trypsin (Promega #V5111)) to facilitate an on-bead protein digest.

This digest was stopped using a protocol modified from *Prikas et al., 2020*. Beads were placed on a magnetic rack (NEB, #S1506S) so the supernatant could be transferred to new 1.5 mL tubes without the beads; peptides from digested proteins in the supernatant will henceforth be referred to as 'unbound' samples, whereas peptides attached to beads will be called 'bound' samples. For 'unbound samples', trifluoroacetic acid (TFA) was added to each sample to a final concentration of 0.1% (w/v), centrifuged (20,000 rcf, 22 °C, 20 min), and the supernatant transferred to new 1.5 mL tubes. To ensure as much protein as possible was recovered, 0.1% (w/v) of TFA was then added to the previously centrifuged 1.5 mL tubes, re-centrifuged (20,000 rcf, 22 °C, 10 min), and the resulting supernatant pooled with the first supernatant sample. For 'bound' samples, magnetic beads were first gently agitated on a tube rotator in 250 µL of elution buffer (EB) solution (80% (v/v) acetonitrile, 0.2% (w/v) TFA, 0.1% (v/v) formic acid), magnetised so the supernatant could be transferred, and then boiled in 200 µL of EB solution at 800 rpm, 95 °C for 5 min. These beads were then magnetised again so that the supernatant could be transferred for TFA treatment.

Peptides for both 'unbound' and 'bound' samples were desalted using tC18 cartridges (Waters, #WAT036810), vacuum-dried at ambient temperature for ~3 hr, and then resuspended in a compatible solution (0.2% (v/v) heptafluorobutyric acid in 1% (v/v) formic acid) for liquid chromatography with tandem mass spectrometry (LC-MSMS).

### Mass spectrometry

For technical reasons, we used two mass spectrometers as outlined in *Prikas et al., 2020* – the Thermo Fisher Scientific Q-Exactive Orbitrap (QE) and ThermoScientific Orbitrap Exploris (Exploris). Samples from biological replicates 1 and 2 were run on the QE, replicates 4 and 5 were run on the Exploris, and replicate 3 was run on both machines. We treated these as six separate experiments, although there were only 5 biological replicates, as considerably more proteins were identified using the Exploris compared with the QE – for this reason, the Exploris was used for subsequent experiments. The overlap between learning proteomes for each biological replicate (i.e. proteins unique to 'TurboID, trained') has been summarised in *Figure 1—figure supplement 4*, based on the mass spectrometer used. The resulting data was processed using the MASCOT search engine (Matrix Science) and *C. elegans* Swiss-Prot database (downloaded on 01/02/2021). An MS/MS ion search was performed with the following settings: 'semi-trypsin' enzyme, 'monoisotopic' mass values, 'unrestricted' protein mass, '±5 ppm' peptide tolerance, '±0.05 Da' fragment mass tolerance, and '3' maximum missed cleavages. Biotin (K), Carbamidomethyl (C), Oxidation (M), Phospho (ST), and Phospho (Y) were selected as variable modifications. Data from the MASCOT search engine is accessible through the Dryad platform (see 'Data Availability').

## Data analysis

We designated protein identities from all mass spectrometry experiments as either 'assigned' or 'unassigned' hits (see *Supplementary file 1C and D* for full lists). 'Assigned proteins/hits' are defined as proteins identified by MASCOT, based on peptide sequences detected during mass spectrometry, with at least one unique peptide detected for that protein. This threshold was chosen based on *Prikas et al., 2020*, to ensure sensitivity in detecting low-abundance neuronal proteins. We did not restrict our definition of 'assigned hits' to any peptide or protein score threshold. In contrast, 'unassigned hits' were determined using peptide sequences with a peptide score ≥15, but that were not assigned a protein identity by MASCOT, as the peptide was not detected as unique for a specific protein by MASCOT. Our criteria for 'unassigned hits' were the protein identity required (1) at least one peptide for the protein with a peptide score ≥15 calculated by MASCOT, (2) a 100% identity match between peptide sequences with a peptide score ≥15 and the protein determined by BLAST, and (3) an e-value <0.05 for the identity match calculated by BLAST, such that a smaller number represents an increased probability that the identity is true and not given by random chance. To collect a list of unassigned hits, we used a custom Python script to perform bulk BLAST-p searches for these sequences using the 'Reference proteins (refseq_protein)' database and '*Caenorhabditis elegans* (taxid:6239)' organism. This Python code uses *Anaconda Software Distribution, 2016*.03 (*Anaconda Software Distribution, 2016*), BioPython 1.78 (*Cock et al., 2009*), and pandas software 1.5.3 (*The pandas development team, 2020*). Our pipeline then assigned protein accession numbers to searched peptide sequences where the percent identity = 100% and e value <0.05. Finally, the Batch Entrez online software available on the NCBI website was used to convert accession numbers to protein identities. This generated lists of unassigned hits for samples in each biological replicate.

Proteins detected in TurboID, trained worms only were calculated as follows: (1) proteins detected from 'non-transgenic, control' and 'non-transgenic, trained' worms were subtracted from corresponding protein lists generated from TurboID worms, and then (2) proteins that overlap between lists for 'TurboID, control' and 'TurboID, trained' worms were subtracted from each other. Venn diagrams were generated using an online tool (https://bioinformatics.psb.ugent.be/webtools/Venn/).

GO term analyses were achieved using STRING (version 12.0), Cytoscape (version 3.10.0), and the Cytoscape App ClueGO (version 2.5.10) (*Bindea et al., 2009*). All 1010 proteins from 'TurboID, trained' lists were entered into STRING as a single list, so k-means clustering could be utilised to separate proteins into 10 clusters, to parse data into more accessible clusters with enough proteins to output enriched GO terms. The 'tabular text' protein-protein interaction information exported from STRING for each cluster was then uploaded onto Cytoscape. GO term analyses were performed with ClueGO for the following categories per cluster using default settings (network specificity in brackets): (i) cellular component (medium), (ii) biological process (detailed), and (iii) molecular function (medium). The ClueGO results were exported to spreadsheets for each cluster, as *Supplementary file 1E* (cellular component) and *Supplementary file 1F* (biological process/molecular function), such that each row corresponds to a GO term based on gene/s within a specific cluster. Each list of genes in each row was consolidated with other gene lists with a matching or similar corresponding GO term, to generate the data shown in *Supplementary file 1E-1F*, *Figure 2*, and *Figure 2—figure supplement 1*. Nodes that did not show protein-protein interactions with other nodes and/or were not categorised into any GO term by ClueGO were manually categorised through a literature search and added onto these figures through Cytoscape.

## Behavioural assays

We adapted previously established methods to perform two behavioural paradigms that model associative learning: (1) salt associative learning (*Tang et al., 2023*; *Hiroki et al., 2022*; *Nagashima et al., 2019*), and (2) salt aversive learning (*Lim et al., 2018*).

### Salt associative learning

This experiment had three groups: naïve worms that did not undergo training, 'control' worms that were paired with 100 mM NaCl and food, and 'trained' worms that were paired with no NaCl and food. Worms from all groups were washed off agar plates using washing buffer as in the 'Protein extraction and quantification' section above. The naïve group could then be immediately tested for their innate response to salt using the salt chemotaxis assay (*Rahmani et al., 2024*). For experimental

groups, worms were washed a third time with washing buffer (containing 50 mM NaCl; for 'control') or no-salt buffer (washing buffer without NaCl; for 'trained'). These washing steps were completed within 10 min per group. Trained animals were placed on 9 cm 'conditioning plates' containing salt-deficient agar (5 mM $K_3PO_4$ (pH 6.0), 1 mM $CaCl_2$, 1 mM $MgSO_4$, 2.0% (w/v) agar) and their bacterial food source *E. coli* MG1655 *bioB::kan*. Control worms were placed on conditioning plates containing salt-deficient agar supplemented with 100 mM NaCl. Worms were left on these plates for 6 hr at 22 °C in the dark (**Nagashima et al., 2019**). Following the training step, worms were washed twice with washing buffer and then transferred within 2 min to test their learning capacity using a salt chemotaxis assay (described below).

The food source (*E. coli* MG1655 *bioB::kan*) was prepared 3–4 days before each experiment, which involved (1) pelleting 1.5 mL of bacteria by centrifugation at 11,000 rpm for 30 s in a 2 mL tube, (2) discarding the supernatant, and then (3) vortexing cells in 750 μL of modified LB. 750 μL of washed bacteria was transferred onto each conditioning plate, left to dry at room temperature overnight, and then left at 22 °C for 3–4 days before use. Notably, the salt-deficient agar in conditioning plates for training contains 0.728 mM of NaCl due to the use of modified LB, meaning that it is not completely lacking NaCl but has only a very small amount.

## Salt aversive learning

This assay involves three groups: naïve (did not undergo training), mock-conditioned worms (paired with no salt and no food), and conditioned worms (paired with 50 mM NaCl and no food). All groups were first washed as described in the 'Salt associative learning' section, except the third wash was performed with no-salt buffer (for 'mock-conditioned') or washing buffer (for 'conditioned'). Naïve worms were placed on chemotaxis assay plates after two washes with washing buffer. Worms undergoing conditioning were incubated at room temperature for 3 hr in 1.5 mL tubes, placed on a shaker at 175 rpm. These tubes contained no-salt buffer for 'mock-conditioned' groups or washing buffer (containing 50 mM NaCl) for 'conditioned' groups. Worms were pelleted by sedimentation for 1–2 min before use in salt chemotaxis assays as described below.

## Salt chemotaxis assay

Salt chemotaxis assay (CTX) plates contain a salt gradient prepared by placing 5 mm cubes of salt-deficient agar on top of salt-deficient agar, with one cube containing 0 mM salt on one side and another cube containing 200 mM salt on the other side (**Jang et al., 2019**) (see **Figure 1—figure supplement 2A** for a schematic). Worms were allowed to crawl freely on chemotaxis assay plates for 45 min in the dark at 22 °C (**Nagashima et al., 2019**), becoming immobilised when they encountered the paralytic agent sodium azide at the extremes of the salt gradient. These animals were then counted within the regions outlined in **Figure 1—figure supplement 2B** to calculate the salt chemotaxis index (CI) based on the below equation:

$$\frac{\# \, worms \, on \, high \, salt \, - \, \# \, worms \, on \, low \, salt}{\# \, total \, population \, - \, \# \, worms \, on \, origin}$$

CI values range from –1.0 (strong preference for low salt concentrations) to +1.0 (strong preference for high salt concentrations). We note several differences in our chemotaxis assay compared with other studies: in our study, (1) CTX plates contained 0 mM salt prior to the addition of salt cubes (containing 0 or 200 mM salt) (**Jang et al., 2019**), while other studies use CTX plates containing 50 mM NaCl (**Kunitomo et al., 2013**), and (2) the food source used to induce learning is biotin-auxotrophic strain (*E. coli bioB::kan* MG1655), which is grown with 50 mM Kanamycin antibiotic, differing from other studies that used *E. coli* NA22 without antibiotic (**Kunitomo et al., 2013**; **Nagashima et al., 2019**).

TurboID worm populations exposed to biotin during training by salt associative learning were scored for the total percentage of transgenic animals on chemotaxis assay plates (**Figure 1B**). This is because TurboID worms expressed the enzyme from an extrachromosomal array, and this scoring confirmed the presence of TurboID-positive animals in worm pellets to be used for downstream proteomics. To do this, fluorescing worms only (identified by co-injection marker P*unc-122::rfp*) were counted in all zones excluding the origin and compared to the total number of worms (fluorescing and non-fluorescing worms) in these zones. The average percentage for TurboID-positive worms was 27–57% per biological replicate.

### Neuron class analysis for learning proteome data

Analyses utilised protein lists (containing assigned hits only) from mass spectrometry experiments for the following groups: (1) non-Tg high-salt control, (2) non-Tg trained, (3) TurboID high-salt control, and (4) TurboID high-salt trained (*Figure 1A*). Briefly, protein identities from non-Tg high-salt control and non-Tg trained were subtracted from TurboID high-salt control and TurboID trained, respectively. Following subtractions, protein lists were then compared to identify those unique to TurboID high-salt control (388 proteins) versus TurboID trained (706 proteins) as in *Figure 1D*.

#### Using CeNGEN

Protein lists unique to TurboID high-salt control and TurboID trained were separately input into the CeNGEN database (threshold = 2) (*Taylor et al., 2021*), to identify gene expression profiles for each list. CeNGEN output 128 lists corresponding to individual neuron classes for each experimental group, and each list contained genes/proteins from one (of two) group/s that express in a specific neuron class. The number of proteins expressed in each neuron class was calculated for TurboID high-salt control and TurboID trained. For TurboID trained, these numbers were normalised by a factor of ~1.8. Neuron-specific fold-differences in the number of proteins expressed in TurboID trained versus TurboID high-salt control were used to rank each neuron class. We interpreted a higher fold-difference value as a relatively greater enrichment (fold-change) of training-associated genes compared to control.

#### Using Worm-Seq and CeSTAAN

Analysis of neuron class using transcriptomics data from Worm-Seq (*Ghaddar et al., 2023*) and CeSTAAN (*Princeton University, 2025*) was performed by (1) downloading gene lists for each neuron class (threshold = 2 for CeSTAAN, no threshold for Worm-Seq as this option was not available), (2) determining the overlap between each neuron class list and assigned hits unique to high-salt control vs trained (from *Figure 1D*), and then (3) calculating the fold-change to rank neuron classes as described with CeNGEN (with normalised values for trained as above).

### Statistical analyses

For behavioural experiments, we performed three to five biological replicates for most genotypes, consistent with similar high-quality studies (*Kitazono et al., 2017*; *Lim et al., 2018*; *Sakai et al., 2017*; *Stein and Murphy, 2014*). For key candidates (KIN-2, F46H5.3, ACC-1, ACC-3, LGC-46), 5 biological replicates were performed. The number of biological replicates is indicated in each figure legend; each biological replicate comprised three technical replicates. Randomisation was not applied because experimental groups were defined by genotype or condition. Quantification of chemotaxis assays was conducted without blinding to genotype or condition. Exclusion criteria were pre-determined: Groups were excluded only if bacterial contamination was evident or if fewer than 20 individual animals were present in a technical replicate.

Statistical analyses were performed in GraphPad Prism (version 8.0). The Shapiro-Wilk normality test was used to assess chemotaxis assay data. Following confirmation of normality, an ordinary two-way ANOVA with Tukey's multiple comparisons post-test ($\alpha$=0.05) was performed to compare differences between mean CI values for each group. This statistical analysis was chosen given that it aligns with recent publications that employ similar experimental designs and data structures (*Beets et al., 2020*; *Jang et al., 2019*; *Kitazono et al., 2017*; *Lim et al., 2018*; *Lin et al., 2010*). Exact p-values for each statistical comparison are reported in *Supplementary file 1I*.

## Acknowledgements

Many thanks go to our colleagues in the Worm Neuroscience lab (Flinders) for thoughtful discussions, and to our colleagues at Flinders University – A/Prof Arne Ittner, Prof Karin Nordström, Dr Alyce Martin, and Dr Amy Wyatt – for feedback, sharing reagents, and scientific advice. We are grateful to the members of the aiLab (Flinders), particularly Dr Emmanuel Prikas, for expert technical advice and assistance. Mass spectrometry data were obtained in the Bioanalytical Mass Spectrometry Facility of the University of New South Wales. We sincerely thank Prof John E Cronan (University of Illinois, USA) for providing *E. coli* strain MG1655. We gratefully acknowledge the

*Caenorhabditis* Genetics Centre, which is supported by the National Institutes of Health (P40 OD010440), and the National Bioresource Project (*C. elegans*) Japan, for providing many of the strains used in this study. AR is funded by a PhD scholarship supported through the Australian Graduate Research Training Program (Flinders University). AM is funded by a Flinders University Research Scholarship (Flinders University). YLC is funded by the National Health and Medical Research Council (NHMRC) (GNT1173448), the Australian Research Council (DP220102511), the Flinders University Parental Leave Research Support Scheme, the Flinders University Impact Seed Funding Grant for Early Career Researchers, and the Flinders Foundation Mary Overton Senior Research Fellowship.

## Additional information

### Funding

| Funder | Grant reference number | Author |
|---|---|---|
| National Health and Medical Research Council | GNT1173448 | Yee Lian Chew |
| Australian Research Council | DP220102511 | Yee Lian Chew |
| Australian Graduate Research Training Program (Flinders University) | | Aelon Rahmani |
| Flinders University Research Scholarship (Flinders University) | | Anna McMillen |

The funders had no role in study design, data collection and interpretation, or the decision to submit the work for publication.

### Author contributions

Aelon Rahmani, Conceptualization, Data curation, Formal analysis, Investigation, Visualization, Methodology, Writing – original draft, Writing – review and editing; Anna McMillen, Radwan Ansaar, Renee Green, Data curation, Formal analysis, Investigation; Ericka Allen, Data curation, Software, Formal analysis, Investigation; Michaela E Johnson, Data curation, Formal analysis, Investigation, Visualization, Writing – review and editing; Anne Poljak, Conceptualization, Data curation, Formal analysis, Supervision, Investigation, Methodology, Project administration, Writing – review and editing; Yee Lian Chew, Conceptualization, Data curation, Formal analysis, Supervision, Funding acquisition, Investigation, Visualization, Methodology, Writing – original draft, Project administration, Writing – review and editing

### Author ORCIDs

Aelon Rahmani https://orcid.org/0009-0007-8498-6901
Anna McMillen https://orcid.org/0009-0008-9655-3350
Ericka Allen https://orcid.org/0009-0001-4749-4509
Radwan Ansaar https://orcid.org/0009-0009-7429-3321
Renee Green https://orcid.org/0009-0001-8287-1955
Michaela E Johnson https://orcid.org/0000-0003-2653-3078
Anne Poljak https://orcid.org/0000-0001-9953-1984
Yee Lian Chew https://orcid.org/0000-0001-6078-9312

Reviewer #1 (Public review): https://doi.org/10.7554/eLife.108438.3.sa1
Reviewer #2 (Public review): https://doi.org/10.7554/eLife.108438.3.sa2
Reviewer #3 (Public review): https://doi.org/10.7554/eLife.108438.3.sa3
Author response https://doi.org/10.7554/eLife.108438.3.sa4

## Additional files

### Supplementary files

Supplementary file 1. Supplementary information for data presented in *Figures 1-7*. (A) List of *C. elegans* lines used in this study. Lines are marked as 'not applicable' for 'number of times backcrossed with N2' when (A) the line itself is N2 (i.e., wild-type) or (B) the strains correspond to transgenic animals with a wild-type background. (B) Quantifying biotinylated protein levels between non-transgenic animals and TurboID *C. elegans*. This excel sheet contains six tables, each corresponding to an individual western blot. These blots are *Figure 1C* (red, rows 1–8), *Figure 1—figure supplement 1* (yellow, rows 9–16), & *Figure 1—figure supplement 3* (green, rows 17–47). Each table compares biotinylated protein signal between experimental groups, which are non-transgenic/Non-Tg (from N2) or TurboID/TbID-expressors (from YLC207). Rows corresponding to *Figure 1C* & *Figure 1—figure supplement 3* involve high-salt control and trained groups prepared for mass spectrometry experiments (i.e., with biotin exposure limited to memory encoding), whereas *Figure 1—figure supplement 1* compares untreated versus biotin-treated animals, Each row corresponds to a unique biological replicate of western blot and mass spectrometry data as annotated under the 'Replicate' column. For each sample, the sum of areas containing bands within an entire lane is used as a readout for biotin-tagged protein signal intensity from whole worm lysates. Non-transgenic/Non-Tg *C. elegans* and TurboID/TbID were treated with biotin during salt associative learning, to generate high-salt control and trained groups for each line. (C) Protein lists for all assigned hits detected in TurboID, trained worms. *C. elegans* peptides were assigned corresponding protein identities by MASCOT and then assigned protein lists were generated for each biological replicate by subtracting proteins also present in 'non-transgenic, trained' worms and/or in 'TurboID, control' worms within the same replicate. Each protein list in rows 9–378 are defined in this table by its corresponding biological replicate, the mass spectrometer used, and the inclusion of data from unbound and bound peptide samples. Peptide samples generated for biological replicate 3 were run in both mass spectrometers used in this study, the Q Exactive or the Exploris, and the corresponding protein list is named '3 a' and '3b' respectively. (D) Protein lists for all unassigned hits detected in TurboID, trained worms. Protein identities were determined through bulk BLAST searching *C. elegans* peptides detected by mass spectrometry, rather than MASCOT, and then unassigned protein lists were generated for each experimental group in each biological replicate. Proteins also in 'non-transgenic, trained' and/or 'TurboID, control' lists for the same biological replicate were subtracted from the corresponding 'TurboID, trained' list. Each protein list in rows 9–587 are defined in this table by its corresponding biological replicate, the mass spectrometer used, and the inclusion of data from unbound and bound peptide samples. Peptide samples generated for biological replicate 3 were run in both mass spectrometers used in this study, the Q Exactive or the Exploris, and the corresponding protein list is named '3 a' and '3b' respectively. (E) Cellular component GO terms associated with assigned hits in the learning proteome. All (1010) assigned protein hits detected in TurboID, trained worms were separated into 10 clusters by k-means clustering on STRING (version 12.0) and GO term analysis was conducted for each cluster using ClueGO (version 2.5.10) on Cytoscape (version 3.10.0). GO terms were then sorted into 'plasma membrane & extracellular matrix (ECM)', 'cilia & dendrites', 'cell body & cytoplasm', 'nucleus', 'Golgi apparatus', 'cytoskeleton', 'mitochondria', 'axon & vesicles', 'pre-synapse', and 'other' (i.e., non-informative terms) categories due to recurring and/or similar terms. Colour-coding is based on the GO Group. Each category is separated by underlined and bolded titles corresponding to the cellular component in rows 1–285. Consolidated lists of genes/proteins are detailed from rows 287–465. (F) Biological process and molecular function GO terms associated with assigned hits in the learning proteome. All (1010) assigned protein hits detected in TurboID, trained worms were separated into 10 clusters by k-means clustering on STRING (version 12.0) and GO term analysis was conducted for each cluster using ClueGO (version 2.5.10) on Cytoscape (version 3.10.0). GO terms were then sorted into 'protein synthesis', 'protein degradation', 'insulin & protein kinases', 'G protein signalling', and 'neurotransmission' categories due to recurring and/or similar terms. Colour-coding is based on the GO Group. Each category is separated by underlined and bolded titles corresponding to the biological processes listed above in rows 1–109. Consolidated lists of genes/proteins are detailed from rows 111–162. (G) Identifying neuron classes represented within the learning proteome in this study. This table presents neurons that express ≥1 gene encoding proteins found in either the trained or control lists. For neurons present in both datasets, fold-change values were calculated as the ratio of the number of genes expressed in the trained condition to those in the control. Neurons are ranked in descending order of fold-change. Values under the '#Proteins from trained (normalised)' column are normalised based on the fold-

difference between the number of proteins in the trained vs control protein lists (normalisation factor = ~1.8). Classifications for neuron type are assigned to each neuron class based on *Pereira et al., 2015*. The CeNGEN database was used to identify gene expression patterns *Taylor et al., 2021* and includes all 302 neurons in the *C. elegans* hermaphrodite nervous system. Only 128 neuron classes are listed here since neurons that do not express any proteins from trained or control groups are not shown. (H) Molecular pathways for proteins detected in TurboID, trained *C. elegans* only. Proteins unique to trained animals that express TurboID were identified by subtracting those also in 'TurboID, control' and/or 'non-transgenic, trained' protein lists. Gene names are listed with assigned hits in bold. Each gene/protein has been grouped based on 'Biological Process' GO term annotations provided by STRING (version 12.0) and information available on WormBase (version WS290) based on data accessed during September-November 2023. (I) Statistical analyses for learning assays performed in this study. Data from statistical analyses are separated by figure with appropriate underlined and bolded titles, including *Figure 1B* (red, rows 1–7), *Figure 4* (yellow, rows 9–31), *Figure 5* (green, rows 33–55), *Figure 6* (blue, rows 57–87), *Figure 6—figure supplement 1* (purple, rows 89–191), *Figure 6—figure supplement 2* (pink, rows 193–247), and *Figure 6—figure supplement 2* (red, rows 249–287). Statistical analyses were performed on GraphPad PRISM 8 using two-way ANOVA and Tukey's multiple comparisons tests (alpha value = $p < 0.05$). Columns left-to-right list the groups being compared, the mean difference (Diff.), the 95% confidence interval of diff., whether the difference is significant, the degree of significance (****≤0.0001; ***≤0.001; **≤0.01; *≤0.05; ns = non-significant), and the adjusted p value.

MDAR checklist

Source data 1. This folder contains information for plasmids pYLC032 and pYLC087, 1742 used in this work to generate new transgenic lines. These plasmids encode *Prab*-1743 *3::V5::TurboID::gpd-2 3' UTR* and *Prab-3::kin-2(ce179)::SL2::tag-RFP::gpd-2 3' UTR*, 1744 respectively. The folder contains an image of each plasmid map, as well as their DNA sequences 1745 in.fasta and.gb formats.

## Data availability

The raw data from this publication, including mass spectrometry data (files from MASCOT search) and raw data from learning assays have been uploaded to Dryad at https://doi.org/10.5061/dryad. 1c59zw43k or included as Source Data (raw western blots, plasmid sequences/maps). Custom Python code is available through GitHub (https://github.com/ChewWormLab/Chew-Worm-Lab-Post-Mass-Spectrometry-Peptide-processing copy archived at *Allen, 2023*). *C. elegans* strains used in this study are available upon request.

The following dataset was generated:

| Author(s) | Year | Dataset title | Dataset URL | Database and Identifier |
|---|---|---|---|---|
| Rahmani A, McMillen A, Allen E | 2025 | Data for: Identifying regulators of associative learning using a protein-labelling approach in *C. elegans* | https://doi.org/10.5061/dryad.1c59zw43k | Dryad Digital Repository, 10.5061/dryad.1c59zw43k |

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
