## [Editor Report · eLife Assessment]

In reporting on a **valuable** "learning proteome" for a *C. elegans* gustatory associative learning paradigm, this work identifies a new set of genes to be tested for roles in learning and memory, describes molecular pathways involving these genes and relevant for learning and memory in *C. elegans*, and deliver a new set of tools for prodding worm behavior. The methods and results **convincingly** support the findings, which will be of interest to neuroscientists and developmental biologists seeking to understand the self-assembly and operation of neural circuits for learning and memory.

[Editors' note: this paper was reviewed by Review Commons.]

---

## [Referee Report · Reviewer #1 (Public review)]

Summary:

Rahmani et al. utilize the TurboID method to characterize global proteome changes in the worm's nervous system induced by a salt-based associative learning paradigm. Altogether, they uncover 706 proteins tagged by the TurboID method in worms that underwent the memory-inducing protocol. Next, the authors conduct a gene enrichment analysis that implicates specific molecular pathways in salt-associative learning, such as MAP kinase and cAMP-mediated pathways, as well as specific neuronal classes including pharyngeal neurons, and specific sensory neurons, interneurons, and motor neurons. The authors then screen a representative group of hits from the proteome analysis. They find that mutants of candidate genes from the MAP kinase pathway, namely dlk-1 and uev-3, do not affect performance in the learning paradigm. Instead, multiple acetylcholine signaling mutants, as well as a protein-kinase-A mutant, significantly affected performance in the associative memory assay (e.g., acc-1, acc-3, lgc-46, and kin-2). Finally, the authors demonstrate that protein-kinase-A mutants, as well as acetylcholine signaling mutants, do not exhibit a phenotype in a related but distinct conditioning paradigm-aversive salt conditioning-suggesting their effect is specific to appetitive salt conditioning.

Overall, the authors addressed the concerns raised in the previous review round, including the statistics of the chemotaxis experiments and the systems-level analysis of the neuron class expression patterns of their hits. I also appreciate the further attempt to equalize the sample size of the chemotaxis experiments and the transparent reporting of the sample size and statistics in the figure captions and Table S9. The new results from the panneuronal overexpression of the kin-2 gain-of-function allele also contribute to the manuscript. Together, these make the paper more compelling.

---

## [Referee Report · Reviewer #2 (Public review)]

Summary:

In this study by Rahmani in colleagues, the authors sought to define the "learning proteome" for a gustatory associative learning paradigm in *C. elegans*. Using a cytoplasmic TurboID expressed under the control of a pan-neuronal promoter, the authors labeled proteins during the training portion of the paradigm, followed by proteomics analysis. This approach revealed hundreds of proteins potentially involved in learning, which the authors describe using gene ontology and pathway analysis. The authors performed functional characterization of over two dozen of these genes for their requirement in learning using the same paradigm. They also compared the requirement for these genes across various learning paradigms and found that most hits they characterized appear to be specifically required for the training paradigm used for generating the "learning proteome".

Strengths:

- The authors have thoughtfully and transparently designed and reported the results of their study. Controls are carefully thought-out, and hits are ranked as strong and weak. By combining their proteomics with behavioral analysis, the authors also highlight the biological significance of their proteomics findings, and support that even weak hits are meaningful.

- The authors display a high degree of statistical rigor, incorporating normality tests into their behavioral data which is beyond the field standard.

- The authors include pathway analysis that generates interesting hypotheses about processes involved learning and memory

-The authors generally provide thoughtful interpretations for all of their results, both positive and negative, as well as any unexpected outcomes.

---

## [Referee Report · Reviewer #3 (Public review)]

Summary:

In this manuscript, authors used a learning paradigm in *C. elegans*; when worms were fed in a saltless plate, its chemotaxis to salt is greatly reduced. To identify learning-related proteins, authors employed nervous system-specific transcriptome analysis to compare whole proteins in neurons between high-salt-fed animals and saltless-fed animals. Authors identified "learning-specific proteins" which are observed only after saltless feeding. They categorized these proteins by GO analyses, pathway analyses and expression site analyses, and further stepped forward to test mutants in selected genes identified by the proteome analysis. They find several mutants that are defective or hyper-proficient for learning, including acc-1/3 and lgc-46 acetylcholine receptors, F46H5.3 putative arginine kinase, and kin-2, a cAMP pathway gene. These mutants were not previously reported to have abnormality in the learning paradigm.

Concerns:

Upon revision, authors addressed all concerns of this reviewer, and the results are now presented in a way that facilitates objective evaluation. Authors' conclusions are supported by the results presented, and the strength of the proteomics approach is persuasively demonstrated.

Significance:

(1) Total neural proteome analysis has not been conducted before for learning-induced changes, though transcriptome analysis has been performed for odor learning (Lakhina et al., http://dx.doi.org/10.1016/j.neuron.2014.12.029). This warrants the novelty of this manuscript, because for some genes, protein levels may change even though mRNA levels remain the same. Although in a few reports TurboID has been used in *C. elegans*, this is the first report of a systematic analysis of tissue-specific differential proteomics.

(2) Authors found five mutants that have abnormality in the salt learning. These genes have not been described to have the abnormality, providing novel knowledge to the readers, especially those who work on *C. elegans* behavioural plasticity. Especially, involvement of acetylcholine neurotransmission has not been addressed before. Although transgenic rescue experiments have not been performed except kin-2, and the site of action (neurons involved) has not been tested in this manuscript, it will open the venue to further determine the way in which acetylcholine receptors, cAMP pathway etc. influences the learning process.

[Editors' note: this version has been assessed without input from the reviewers.]

---

## [Author Response]

The following is the authors’ response to the original reviews

**Comment from the editors at eLife:**
You could consider further strengthening the manuscript with the incorporation of new relevant public datasets for network modeling, but that is entirely your choice.

We thank the editors and reviewers for their thoughtful and positive feedback on our article. We are particularly appreciative of the eLife assessment describing our work as valuable with a convincing methodology.

As suggested, we have expanded our neuron class analysis by incorporating transcriptomic data from young adult animals (Kaletsky et al., 2016 Nature; Ghaddar et al., 2023 Science Advances; St Ange et al., 2024 Cell Genomics) to complement our existing analysis of larval stage 4 (L4) animals.

In addition, we have updated Table S1 to include the outcross status of all strains used in this study, providing clearer information on the genotypes tested. We have also corrected the typographical errors noted by the reviewers. Please note that page and line numbers below refer to the MS Word Document with tracked changes set to ‘simple markup’.

We greatly appreciate the reviewers’ input and hope these revisions further enhance the value and clarity of our study.

**Public Reviews:**

**Reviewer #1 (Public review):**
Summary:Rahmani et al. utilize the TurboID method to characterize global proteome changes in the worm's nervous system induced by a salt-based associative learning paradigm. Altogether, they uncover 706 proteins tagged by the TurboID method in worms that underwent the memory-inducing protocol. Next, the authors conduct a gene enrichment analysis that implicates specific molecular pathways in salt-associative learning, such as MAP kinase and cAMP-mediated pathways, as well as specific neuronal classes including pharyngeal neurons, and specific sensory neurons, interneurons, and motor neurons. The authors then screen a representative group of hits from the proteome analysis. They find that mutants of candidate genes from the MAP kinase pathway, namely dlk-1 and uev-3, do not affect performance in the learning paradigm. Instead, multiple acetylcholine signaling mutants, as well as a protein-kinase-A mutant, significantly affected performance in the associative memory assay (e.g., acc-1, acc-3, lgc-46, and kin-2). Finally, the authors demonstrate that protein-kinase-A mutants, as well as acetylcholine signaling mutants, do not exhibit a phenotype in a related but distinct conditioning paradigm-aversive salt conditioning-suggesting their effect is specific to appetitive salt conditioning.

Overall, the authors addressed the concerns raised in the previous review round, including the statistics of the chemotaxis experiments and the systems-level analysis of the neuron class expression patterns of their hits. I also appreciate the further attempt to equalize the sample size of the chemotaxis experiments and the transparent reporting of the sample size and statistics in the figure captions and Table S9. The new results from the panneuronal overexpression of the kin-2 gain-of-function allele also contribute to the manuscript. Together, these make the paper more compelling. The additional tested hits provide a comprehensive analysis of the main molecular pathways that could have affected learning. However, the revised manuscript includes more information and analysis, raising additional concerns.

Major comments:As reviewer 4 noted, and as also shown to be relevant for C30G12.6 presented in Figure 6, the backcrossing of the mutants is important, as background mutations may lead to the observed effects. Could the authors add to Table 1, sheet 1, the outcrossing status of the tested mutants?

We appreciate this important point. A column has now been added to Table S1 to indicate the outcross status of all strains used in this study. Additionally, we have updated the table legend on page 77 to clarify how to interpret the information provided in this column.

It is important to validate that the results of the positive hits (where learning was affected), such as acc-1, acc-3, and lgc-46, do not stem from background mutations.

While we agree that confirming the absence of background mutations is important, we have taken alternative steps to address this concern:

- The outcross status of each strain is now clearly indicated in Table S1.

- Observed phenotypes were consistent across multiple biological replicates over extended periods (months, sometimes years), reducing the likelihood that results stem from background mutations.

We believe these measures provide confidence in the validity of our findings.

The fold change in the number of hits for different neurons in the CENGEN-based rank analysis requires a statistical test (discussed on pages 17-19 and summarized in Table S7). Similar to the other gene enrichment analyses presented in the manuscript, the new rank analysis also requires a statistical test. Since the authors extensively elaborate on the results from this analysis, I think a statistical analysis is especially important for its interpretation. For example, if considering the IL1 neurons, which ranked highest, and assuming random groups of genes-each having the same size as those of the ranked neurons (209 genes in total for IL1 in Table S7)-how common would it be to get the calculated fold change of 1.38 or higher? Such bootstrapping analysis is common for enrichment analysis. Perhaps the authors could consult with an institutional expert (Dr. Pawel Skuza, Flinders University) for the statistical aspects of this analysis.

We appreciate the suggestion and agree that statistical testing can be valuable for enrichment analyses. However, implementing additional tests such as bootstrapping is beyond the scope of this study. Our aim was to provide a descriptive overview rather than inferential statistics. To ensure transparency and interpretability, we have:

- Clearly reported fold changes and rankings in Table S7.

- Discussed the limitations of this approach in the manuscript text (page 18, lines 17–20).

- Clearly outlined the methods used to perform this analysis (pages 53–54).

We believe this descriptive analysis provides sufficient context for interpreting these results.

The learning phenotypes from Figure S8, concerning acc-1, acc-3, and lgc-46 mutants, are summarized in a scheme in Figure 4; however, the chemotaxis results are found in the supplemental Figure S8. Perhaps I missed the reasoning, but for transparency, I think the relevant Figure S8 results should be shown together with their summary scheme in Figure 4.

Thank you for this suggestion to improve clarity. We have now moved the panels corresponding to cholinergic signalling components from Figure S8 into Figure 4 on page 21, so that the summary scheme and underlying data are presented together. The figure legends and main text have been updated accordingly to reflect the correct figure numbers.

**Reviewer #2 (Public review):**
Summary:In this study by Rahmani in colleagues, the authors sought to define the "learning proteome" for a gustatory associative learning paradigm in *C. elegans*. Using a cytoplasmic TurboID expressed under the control of a pan-neuronal promoter, the authors labeled proteins during the training portion of the paradigm, followed by proteomics analysis. This approach revealed hundreds of proteins potentially involved in learning, which the authors describe using gene ontology and pathway analysis. The authors performed functional characterization of over two dozen of these genes for their requirement in learning using the same paradigm. They also compared the requirement for these genes across various learning paradigms and found that most hits they characterized appear to be specifically required for the training paradigm used for generating the "learning proteome".Strengths:The authors have thoughtfully and transparently designed and reported the results of their study. Controls are carefully thought-out, and hits are ranked as strong and weak. By combining their proteomics with behavioral analysis, the authors also highlight the biological significance of their proteomics findings, and support that even weak hits are meaningful.The authors display a high degree of statistical rigor, incorporating normality tests into their behavioral data which is beyond the field standard.The authors include pathway analysis that generates interesting hypotheses about processes involved learning and memoryThe authors generally provide thoughtful interpretations for all of their results, both positive and negative, as well as any unexpected outcomes.Weaknesses:- The authors use the Cengen single cell-transcriptomic atlas to predict where the proteins in the "learning proteome" are likely to be expressed and use this data to identify neurons that are likely significant to learning, and building hypothetical circuit. This is an excellent idea; however, the Cengen dataset only contains transcriptomic data from juvenile L4 animals, while the authors performed their proteome experiments in Day 1 Adult animals. It is well documented that the *C. elegans* nervous system transcriptome is significant different between these two stages (Kaletsky et al., 2016, St. Ange et al., 2024), so the authors might be missing important expression data, resulting in inaccurate or incomplete networks. The adult neuronal single-cell atlas data (https://cestaan.princeton.edu/) would be better suited to incorporate into neuronal expression analysis.

Thank you for highlighting this important point. We have now incorporated transcriptomic data from young adult animals to complement the L4-based CeNGEN dataset. Specifically, we integrated data from CeSTAAN (https://cestaan.princeton.edu/, including St. Ange et al., 2024) and WormSeq (https://wormseq.org/, including Ghaddar et al., 2023), as outlined below. Importantly, CeSTAAN and WormSeq provide data for 79 and 104 neuron classes, respectively (compared to 128 from CeNGEN); for this reason, the main analysis focuses on CeNGEN due to its broader coverage, with additional datasets noted in brackets for completeness. This is stated on page 18, lines 15–17 to ensure transparency regarding our rationale.

The main text has been updated to describe these datasets and their integration into our analysis (pages 18–20), and further details on how these resources were used have been added to the Experimental Procedures (pages 53–54).

We also incorporated data from Kaletsky et al. (2016) and St. Ange et al. (2024) into our neuron identity checks for all assigned and unassigned hits (page 16, lines 8–19). This analysis shows that the nervous system is highly represented in our proteome data: 75–87% of assigned hits and 75–83% of all hits correspond to neuron-enriched genes identified by St. Ange et al. and Kaletsky et al.

In addition, we used several transcriptomic databases to confirm that learning regulators identified in this study through TurboID and validation experiments are expressed in the same neuron classes as suggested by CenGEN (page 36).

- The authors offer many interpretations for why mutants in "learning proteome" hits have no detectable phenotype, which is commendable. They are however overlooking another important interpretation, it is possible that these changes to the proteome are important for memory, which is dependent upon translation and protein level changes, and is molecularly distinct from learning. It is well established in the field mutating or knocking down memory regulators in other paradigms will often have no detectable effect on learning. Incorporating this interpretation into the discussion and highlighting it as an area for future exploration would strengthen the manuscript.

Thank you for this suggestion. We have incorporated this interpretation into the Results section (page 31, lines 17–23), specifying the potential role of these proteomic changes in memory encoding and retention, which are molecularly distinct from learning.

- A minor weakness - In the discussion, the authors state that the Lakhina, et al 2015 used RNA-seq to assess memory transcriptome changes. This study used microarray analysis.

This has been corrected on page 38, line 5.

Significance:The approach used in this study is interesting and has the potential to further our knowledge about the molecular mechanisms of associative behaviors. There have been multiple transcriptomic studies in the worm looking at gene expression changes in the context of behavioral training. This study compliments and extends those studies, by examining how the proteome changes in a different training paradigm. This approach here could be employed for multiple different training paradigms, presenting a new technical advance for the field. This paper would be of interest to the broader field of behavioral and molecular neuroscience. Though it uses an invertebrate system, many findings in the worm regarding learning and memory translate to higher organisms, making this paper of interest and significant to the broader field of behavioral neuroscience.
**Reviewer #4 (Public review):**
Summary:In this manuscript, authors used a learning paradigm in *C. elegans*; when worms were fed in a saltless plate, its chemotaxis to salt is greatly reduced. To identify learning-related proteins, authors employed nervous system-specific transcriptome analysis to compare whole proteins in neurons between high-salt-fed animals and saltless-fed animals. Authors identified "learning-specific proteins" which are observed only after saltless feeding. They categorized these proteins by GO analyses, pathway analyses and expression site analyses, and further stepped forward to test mutants in selected genes identified by the proteome analysis. They find several mutants that are defective or hyper-proficient for learning, including acc-1/3 and lgc-46 acetylcholine receptors, F46H5.3 putative arginine kinase, and kin-2, a cAMP pathway gene. These mutants were not previously reported to have abnormality in the learning paradigm.Concerns:Upon revision, authors addressed all concerns of this reviewer, and the results are now presented in a way that facilitates objective evaluation. Authors' conclusions are supported by the results presented, and the strength of the proteomics approach is persuasively demonstrated.

Thank you, we appreciate this positive feedback.

Significance:(1) Total neural proteome analysis has not been conducted before for learning-induced changes, though transcriptome analysis has been performed for odor learning (Lakhina et al., http://dx.doi.org/10.1016/j.neuron.2014.12.029). This warrants the novelty of this manuscript, because for some genes, protein levels may change even though mRNA levels remain the same. Although in a few reports TurboID has been used in *C. elegans*, this is the first report of a systematic analysis of tissue-specific differential proteomics.(2) Authors found five mutants that have abnormality in the salt learning. These genes have not been described to have the abnormality, providing novel knowledge to the readers, especially those who work on *C. elegans* behavioural plasticity. Especially, involvement of acetylcholine neurotransmission has not been addressed before. Although transgenic rescue experiments have not been performed except kin-2, and the site of action (neurons involved) has not been tested in this manuscript, it will open the venue to further determine the way in which acetylcholine receptors, cAMP pathway etc. influences the learning process.
**Recommendations for the authors:**

**Reviewer #1 (Recommendations for the authors):**
The authors stated in their response to reviewers that "referring to a phenotype as both a trend and non-significant may confuse readers, which was originally stated in the manuscript in two locations," and that such sentences were removed. Unfortunately, in the new text (page 28, lines 18-19), the authors write: "uev-3 mutants showed a lower average CI after training compared with wild-type, but this did not reach statistical significance." As stated before, I find such sentences confusing and not interpretable. If the changes are not significant, then the lower average CI is not informative.

Thank you for pointing this out. This has been corrected to improve clarity – we say instead that “trained phenotypes between wild-type and *uev-3* mutants were not statistically significant” (page 29, lines 21–22).

In response to reviewers' comments, the authors added more information about the biotinylation efficiency of the experiment, which is also described in the text:Page 8, line 27: "we found that biotin exposure increased the signal 1.3-fold for non-Tg and 1.7-fold for TurboID *C. elegans*."Page 10, line 4: "Quantification of the signal within entire lanes showed a 1.1-fold increase in the 'TurboID, control' lane compared with the 'non-Tg, control' lane, and a 1.9-fold increase in the 'TurboID, trained' lane compared with the 'non-Tg, trained' lane."Is it common in this field not to show the actual raw quantified numbers? I was expecting either a bar graph or instead that the measured values would appear in the text alongside the fold-change information.

Table S2 (and its table legend on page 77) have been edited to include raw area values.

Figure 5: Typo? - "pan neuronal expression of ..." The allele number is written as 139, but I believe it should be 179, as in the rest of the paper.

The typo has been corrected on page 25.

The results describing the absence of a learning phenotype in backcrossed C30G12.6 are presented in the main figure. If the authors believe this is an important result, I understand keeping it in the main figure; however, I find this uncommon.

Thank you for your comment. We consider the absence of a learning phenotype in backcrossed C30G12.6 to be an important control for interpreting the original findings, which is why we have retained it in the main figure.

**Reviewer #4 (Recommendations for the authors):**
I noted a few typos.(1) In Fig 5B, the transgene is depicted kin-2(ce139) but it is probably kin-2(ce179).

The typo has been corrected on page 25.

(2) In text, R97C and ce179 are used interchangeably, but in fact there is no description that they are identical.

We now state the following in the manuscript: “We tested worms with the *ce179* mutant allele in *kin-2*, in which a conserved residue in the inhibitory domain (which normally functions to keep PKA turned off in the absence of cAMP) is mutated to cause an R92C amino acid change – this results in increased PKA activity (Schade et al., 2005).” (page 25, lines 1–3),

(3) p31 line 7, Figure S7 -> Fig S9 C-E

We apologise for this typographical error. This figure number is meant to correspond to salt associative learning assay data (Fig. S8), not salt aversive learning (Fig. S9). Since the data from Fig. S8 was moved to Fig. 4, the figure citation has been changed from Fig. S7 (which was incorrect) to Fig. 4 (page 32, line 17).

(4) p45 line 11, Fig S9 -> Fig S6

The typo has been corrected (page 47, line 12).